

# Development of the morphodynamics on LIA lateral moraines in ten glacier forefields of the Eastern Alps since the 1950s

Sarah Betz-Nutz[1], Tobias Heckmann[1], Florian Haas[1], Michael Becht[1]

[1]Physical Geography, Catholic University of Eichstätt-Ingolstadt, Eichstätt, 85072, Germany

5  *Correspondence to*: Sarah Betz-Nutz (sarah.betz@ku.de)

**Abstract.**

Since the end of the Little Ice Age (LIA) in the middle of the 19[th] century, Alpine glaciers have been subject to severe recession that is enhanced by the recent global warming. The melting glaciers expose large areas with loose sediments, amongst others in the form of lateral moraines. Due to their instability and high slope angle, the lateral moraines are reworked by 10 geomorphological processes such as debris flows, slides or fluvial erosion. In this study, the development of the morphodynamics and changes of geomorphological processes on lateral moraines were observed over decades, based on a selection of 10 glacier forefields in the eastern Alps. To identify geomorphological changes over time, several datasets of archival aerial images reaching back to the 1950s were utilized in order to generate DEMs and DEMs of difference. The aerial images were complemented by recent drone images for selected moraine sections, enabling a high-resolution analysis of the 15 processes currently occurring. The results concerning the development of morphodynamics on lateral moraine sections are diverse: some slopes display a stagnation of the erosion rates, the rates on one section increase significantly and the majority of the slopes show a decline of the morphodynamics over decades, which, however, stay on a high level in many cases. In particular, moraine sections with high morphodynamics in the beginning of the observation period mostly show high erosion rates up until now with values up to 11 cm per year. These moraine sections also feature heavy gullying on their upper slopes. 20 A correlation between the development of morphodynamics and the time since deglaciation could scarcely be established. In fact, the results rather indicate that characteristics of the lateral moraines such as the initial slope angle at the time of deglaciation have a significant influence on the later morphodynamics. These observations raise concerns whether often conducted analyses based on the comparison of lateral moraine sections with different distances to the glacier terminus, assumed to represent varying time spans since deglaciation, can provide sound evidence concerning their process of 25 stabilization.

## 1 Introduction

Hambrey (1994, p.142) wrote that "lateral moraines are among the most impressive features of contemporary glacial mountain environments […]". These lateral moraines, often showing a perfect, well-preserved shape (Mortara and Chiarle, 2005), are exposed in consequence to the extensive glacier melting since the end of the Little Ice Age (LIA, ca. 1450–1850) (Haeberli 30 and Beniston, 1998; Zemp et al., 2008).



The unconsolidated, unvegetated and often steep proximal moraine slopes are prone to a variety of geomorphic processes (Curry et al., 2009; Micheletti et al., 2015b). Water as well as gravity-driven processes are occurring on the moraines. These include linear processes such as fluvial rill erosion or debris flows, and denudation by slope wash, small mass movements or sheet erosion by snow gliding (cf. Ballantyne, 2002b; Chiarle et al., 2007; Heckmann et al., 2012; Mortara and Chiarle, 2005;

Wetzel, 1992). Debris flows were identified in several studies as the dominant agent of erosion and transport on recently deglaciated sediment-mantled slopes with high slope gradients (Ballantyne, 2002b; Curry et al., 2006). On some of these steep lateral moraines, the geomorphic processes, mainly fluvial processes and in the following also debris flows (cf. Ballantyne and Benn, 1996), lead to the formation of deeply incised gullies (Curry, 1999; Curry et al., 2006). On the other hand, Lukas et al. (2012) point out the unusually-high stability and steepness of some lateral moraines in the Alps due to overconsolidation.

Church & Ryder (1972) and Ballantyne (2002a) developed models predicting the course of the erosion rates in a paraglacial landscape and the duration until stabilization. In these models, "paraglacial" refers to a time period in which non-glacial geomorphic processes lead to an adjustment of bedrock and sedimentary landforms after deglaciation; the term also implies that the activity and dynamics of these processes are directly conditioned by deglaciation (Ballantyne, 2002b, a; Benn and Evans, 1998; Church and Ryder, 1972). Both, Church & Ryder (1972) and Ballantyne (2002a), assume a rapid decrease of the

sediment yield after deglaciation, Ballantyne even claiming an exponential decrease with 50 % of sediment being exhausted after 354 years in his hypothetical model.

Several studies tried to verify these models based on investigations of gullies on LIA-lateral moraines. Curry (1999) and Curry et al. (2006) analysed gullies on moraine sections of different age concerning the time since deglaciation, which have thus undergone different times for paraglacial adjustment. They found out that the older gullies are less narrowly incised, the gully

depth reduces with age and the sidewalls collapse. The authors conclude that this leads to a levelling and finally stabilization of the slopes within decades. Also Ballantyne & Benn (1996) deduced a stabilization from their analyses and Eichel et al. (2018) developed a conceptual model of a paraglacial transition from active to stable Alpine lateral moraine slopes based on their observations.

These analyses, however, did not observe the geomorphic activity on lateral moraines over time. Instead, the observation over

time was substituted by the comparison of moraine sections with different distances to the glacier terminus, thus with different age since deglaciation. This so-called space-for-time substitution approach, also often used in ecology (Damgaard, 2019), however, ignores different characteristics of the moraine sections, e.g. regarding slope angle, slope length etc. Thus, the comparability of the moraine sections is not clear. Moreover, these studies combined the space-for-time substitution with morphometric measurements of gullies for the estimation of the eroded volume (e.g. Curry, 1999; Curry et al., 2006) or

geomorphological mapping for the estimation of the morphodynamics (e.g. Eichel et al., 2018), which means that no direct measurements of erosion were conducted.

So far, only few studies observed lateral moraines over time using multitemporal digital elevation models (DEMs), e.g. Lane et al. (2017) or Dusik et al. (2019). These studies could not confirm a levelling of gullies after some decades as reported from Curry (1999) and Curry et al. (2006), neither could Jäger & Winkler (2012) making a comparison to old photographs (ca.





1950–1990). Only Schiefer & Gilbert (2007) could detect decreasing erosion in gullies over decades in Canada using multitemporal DEMs.

In our study, we use several datasets of historical aerial images dating back to the 1950s, from which multitemporal DEMs were generated. This enables an observation of the morphodynamics on the lateral moraines over decades and the investigation of landform evolution over time on the same moraine section, rendering the space-for-time substitution unnecessary. It also

allows for a validation of the latter. Another advantage of using multitemporal DEMs is the possibility to identify the melt-out of dead ice contained in the lateral moraines which must not be interpreted as erosion (Anderson, 2019). It is unclear if a possible influence of melting dead ice on paraglacial adjustment was considered in the named studies using space-for-time substitution. In this paper, we take account of this aspect by carefully excluding areas of conspicuous melt-out of dead ice from our analyses.

We analyse the development of the morphodynamics on ten selected lateral moraines tracts distributed within the Eastern Alps. This enables a good comparison and covers different characteristics regarding e.g. altitude a.s.l. and climate. The analysis over time using multitemporal DEMs as well as the larger number of glacier forefields furthermore enable the investigation of the possible influence of the time since deglaciation and the slope angle on the development of the morphodynamics and the process of paraglacial adjustment.

## 2  Study areas

Ten glacier forefields in the Eastern Alps were selected for the investigations, located in Germany, Austria and Italy (see Fig. 1a). The glacier forefields and thus the study areas were defined in a way that they encompass the extent of the corresponding glacier during the Little Ice Age (LIA; see e.g. Fischer et al., 2015). They follow the lateral moraine ridges up to the maximum LIA extent. On the upper border perpendicular to the glacier flow direction, the areas were usually delimited in a way that they

include areas still covered by the glacier on the earliest aerial images in the 1950s. This often corresponds to the upper limit of lateral moraines being developed. The study areas are marked in black in Fig. 1b to k.

Due to their different locations within the Eastern Alps, mainly the central Alps and the northern and southern Alps, and the different altitudes between 1927 and 3049 m a.s.l., the study areas feature different climatic properties that are reflected e.g. by differences in the mean annual precipitation and the mean annual air temperature as listed in Table 1.

Besides the climatic conditions, the study areas also show different lithological characteristics. While five study areas belong to the Ötztal-Stubai Complex, two are situated within the Tauern window, one in the Ortler-Campo Crystalline, one in the Adamello-Presanella group and one in the Wetterstein mountains (e.g. Meschede, 2018; Mair et al., 2007; Pindur and Luzian, 2007; Carton and Baroni, 2017; see Table 1).

Resulting from the different climatic and also lithological conditions, the glacier forefields also show different kinds of

vegetation. The majority of the glacier forefields is part of the alpine zone which is dominated by dwarf shrubs and alpine pastures (Kilian et al., 1994; Pindur and Luzian, 2007; Veit, 2002). In some of the lower parts of the glacier forefields, larch- and Swiss stone pine forests can be found (Autonomous Province Bolzano-South Tyrol, 2010). In any case, it has to be





considered that due to the glacier coverage a growth of vegetation is only possible since maximum 170 years and the vegetation within the glacier forefields is less developed than in surrounding areas.

The sizes of the glacier forefields and thus the study areas differ due to different sizes of the glaciers which formed them, varying between 16 and 278 ha (see Table 1). As a consequence, the corresponding lateral moraines have different slope lengths, slope angles and lengths of the moraine tracts.

Within each glacier forefield, we selected between two and three lateral moraine sections, depending on the size of the respective forefield, for closer analysis of the development of the geomorphological processes and for the calculation of erosion

rates on the upper moraine slopes. These moraine sections were chosen so that they are located in different distances from the LIA-maximum extent of the respective glacier, representing different time periods since deglaciation, different altitudes a.s.l., different slope angles etc. Criteria for the selection were mainly a well-developed moraine section without larger bedrock outcrops, an optimum distance between the sections and accessibility. The channel usually present at the foot of the slope and the moraine ridge or gully headcut served as lower and upper borders for the moraine section, respectively. The boundaries on

the left and right side were either defined by geomorphological features such as outcropping bedrock or a channel crossing the area, or they were delineated as watershed boundaries in order to avoid areas with sediment influx from adjacent sections. The moraine sections were named corresponding to the glaciers which formed them, as it can be seen in Fig. 1b to k with the purple, cyan and green polygons and labels. Properties like the slope angle or altitude a.s.l. of each moraine section can be found in Betz-Nutz (2021, p.44).

**Table 1: Size, altitude a.s.l., mean annual precipitation, mean annual temperature and geological unit of the glacier forefields (source of precipitation and temperature data: www.alpenklima.eu, 3PClim-project)**

| Study area | Size (ha) | Altitude (m a.s.l.) | Mean annual precipitation (mm) | Mean annual air temperature (°C) | Geological unit |
|---|---|---|---|---|---|
| **Alpeiner Ferner** | 128 | 2265–2755 | 1251–1500 | -4 to 0 | Ötztal-Stubai-Complex |
| **Gepatschferner** | 278 | 1938–2616 | 1101–1250 | -2 to 2 | Ötztal-Stubai-Complex |
| **Höllentalferner** | 16 | 2115–2353 | 1501–1750 | 0 to 2 | Wetterstein mountains |
| **Hohenferner** | 40 | 2593–2876 | 951–1100 | -4 to 0 | Ortler-Campo Crystalline |
| **Krimmler Kees** | 138 | 1864–2470 | 1501–1750 | -2 to 2 | Tauern Window |
| **Langtauferer Ferner** | 124 | 2064–2668 | 951–1100 | -2 to 2 | Ötztal-Stubai-Complex |
| **Rofenkarferner** | 43 | 2620–3049 | 951–1100 | -6 to -2 | Ötztal-Stubai-Complex |
| **Vedretta d'Amola** | 38 | 2450–2741 | 1251–1500 | -2 to 0 | Adamello-Presanella group |
| **Waxeggkees** | 132 | 1927–2635 | 1501–1750 | -2 to 2 | Tauern Window |
| **Weißseeferner** | 143 | 2361–2946 | 1101–1250 | -4 to 0 | Ötztal-Stubai-Complex |





**Figure 1: Study areas: (a) Map with an overview over the location of the glacier forefields (ASTER DEM of NASA, METI and J-spacesystems), (b) – (k) orthophotos showing the ten investigated glacier forefields and the moraine sections which were investigated in detail, arrows showing the flow direction of the glacier. Orthophotos calculated on base of the following aerial image datasets: Alpeiner Ferner 2009 (Office of the Tyrolean government, Department of Geoinformation, Austria), Langtauferer Ferner 2016 (Hydrographic Office, Agency for Civil Protection, Autonomous Province of Bolzano-South Tyrol, Italy), Krimmler Kees 2006 (Federal Office of Metrology and Surveying BEV, Austria), Vedretta d'Amola 2004 (Autonomous Province of Trento, Italy), Höllentalferner 2009 (Bavarian State Office for Survey and Geoinformation LDBV, Germany), Hohenferner 2016 (Hydrographic Office, Agency for Civil Protection, Autonomous Province of Bolzano-South Tyrol, Italy), Rofenkarferner 2009 (Office of the Tyrolean government, Department of Geoinformation, Austria), Waxeggkees 2010 (Office of the Tyrolean government, Department of Geoinformation, Austria); Gepatschferner and Weißseeferner orthophoto of 2015 (State of Tyrol, Austria)**



## 3 Data and Methods

### 3.1 Data

For the analysis in this study, different kinds of data were used. An overview of the data including the sources is listed in Table 2. Aerial images formed the basis for the calculation of DEMs and orthophotos for the ten entire glacier forefields (study areas). For most of the study areas, three aerial image datasets were available, mostly one from the 1950s, one from the 1970s/1980s and one from the 2000s. The number of the aerial images amounts to five to eight images for the older datasets (until 1983) which were scanned, digitized images, and to five to 30 images for the more recent, digital image datasets (from the 2000s on). DEMs from airborne laserscanning (ALS) surveys were used in addition, or to close gaps in the photogrammetric DEMs (see Table 2).

Moreover, drone images were taken of the 23 selected moraine sections for two consecutive years between 2017 and 2019. These were also the basis for the calculation of DEMs and orthophotos. For the moraine sections of two study areas, terrestrial laserscanning (TLS) data was used instead of drone images (see Table 2).

Besides the orthophotos generated from the named aerial image datasets, we also used old maps showing the former glacier extent as well as descriptions and mapping in the literature for mapping glacier extents (see section 3.6.1). A complete list of the orthophotos, maps and literature used can be found in Betz-Nutz (2021, p.188–190).

**Table 2: Overview of the used aerial image datasets, ALS- and TLS datasets and drone images with the respective years, sources and study areas. Aerial images and ALS data are covering the entire glacier forefields, whereas the TLS data and drone images cover only the selected moraine sections.**

| Year | Type | Source | Study area |
|---|---|---|---|
| **1953/1954** | Aerial images | Federal Office of Metrology and Surveying (BEV), Austria | Alpeiner Ferner, Gepatschferner, Krimmler Kees, Waxeggkees, Weißseeferner |
| **1959** | Aerial images | Italian Military Geographic Institute (IGMI), Italy | Vedretta d'Amola, Hohenferner, Langtauferer Ferner |
| **1960** | Aerial images | Bavarian State Office for Survey and Geoinformation (LDBV), Germany | Höllentalferner |
| **1970/1971/ 1973/1974** | Aerial images | Office of the Tyrolean government, Department of Geoinformation, Austria | Alpeiner Ferner, Gepatschferner, Krimmler Kees, Rofenkarferner, Waxeggkees, Weißseeferner |
| **1983** | Aerial images | Bavarian State Office for Survey and Geoinformation (LDBV), Germany | Höllentalferner |
| **1983** | Aerial images | Autonomous Province of Trento, Italy | Vedretta d'Amola |
| **2004** | Aerial images | Autonomous Province of Trento, Italy | Vedretta d'Amola |
| **2006** | Aerial images | Federal Office of Metrology and Surveying (BEV), Austria | Krimmler Kees |





| 2006 | ALS data | Autonomous Province of Bolzano-South Tyrol, Italy | Hohenferner, Langtauferer Ferner |
|---|---|---|---|
| 2009 | Aerial images | Bavarian State Office for Survey and Geoinformation (LDBV), Germany | Höllentalferner |
| 2009 | Aerial images | Office of the Tyrolean government, Department of Geoinformation, Austria | Alpeiner Ferner, Rofenkarferner |
| 2010 | Aerial images | Office of the Tyrolean government, Department of Geoinformation, Austria | Waxeggkees |
| 2014/2015 | TLS data | Project PROSA, Cath. Univ. of Eichstätt-Ingolstadt (Dr. Jana-Marie Rabe née Dusik) | Gepatschferner, Weißseeferner |
| 2016 | Aerial images | Hydrographic Office, Agency for Civil Protection, Autonomous Province of Bolzano-South Tyrol, Italy | Hohenferner, Langtauferer Ferner |
| 2017 | ALS data | Project PROSA, Cath. Univ. of Eichstätt-Ingolstadt | Gepatschferner, Weißseeferner |
| 2017 | Drone campaign | Own data | Vedretta d'Amola, Hohenferner, Krimmler Kees, Rofenkarferner, Waxeggkees |
| 2018 | Drone campaign | Own data | Vedretta d'Amola, Alpeiner Ferner, Hohenferner, Höllentalferner, Krimmler Kees, Langtauferer Ferner, Rofenkarferner, Waxeggkees |
| 2019 | Drone campaign | Own data | Alpeiner Ferner, Höllentalferner, Langtauferer Ferner |
| 2019 | ALS data | Project SEHAG, Cath. Univ. of Eichstätt-Ingolstadt | Hohenferner |

## 3.2 Field survey

The field campaigns were conducted between 04.07.2017 and 04.09.2019, each year within the summer months July, August

and September. For global referencing, we defined and measured Ground Control Points (GCPs) on the selected moraine sections of which drone images were taken later. As GCPs, big rocks equally distributed on the moraine slopes were marked with crosses. However, on the steepest slopes, no GCPs could be set on the upper slope parts due to inaccessibility. Yet, on many moraine sections it was possible to access the ridge of the moraine and mark and measure GCPs there. On each moraine section, between 14 and 79 GCPs were marked and measured, depending on the size and terrain complexity of the slope.

The crosses marked on the stones were then surveyed with a total station of Leica Geosystems AG, TCRM1205 Type GDF121. As these measurements were conducted in the local coordinate system of the total station, at least three fix points around the total station were also measured using a Stonex S9III Plus Global Navigation Satellite System (GNSS) antenna for about 1.5



hours per point to gather global coordinates. The recorded raw data was then postprocessed by the Trimble CenterPoint RTX Post-Processing service. The local coordinates of the total station were then rotated into the geocentric coordinate system using

the fix points and a coordinate transformation based on Euler angles. Finally, the GCPs were converted into the UTM coordinate system (Zone 32N, EPSG 25832). In the following years of field campaigns, the already marked and measured GCPs were used again, where they were still available. However, most GCPs and some fix points were measured again as they were not found again or had possibly moved.

After marking and measuring the GCPs, aerial photos of the moraine sections were taken by a DJI Phantom 4 Pro⁺ drone. Care

was taken to achieve a high overlap of the images in horizontal and vertical direction to enable a good photogrammetric processing. The flight strips were parallel to the slope and it was attempted to keep the distance to the slope constant, so that the marked crosses on the stones could be seen well. The mounted camera of the drone was positioned in that way that view direction was orthogonal to the slope surface and for that purpose it was also adapted during the flight. This means it had to be adjusted corresponding to the slope angle which is varying on the lateral moraines from the foot to the ridge. Between 396

and 1610 images were taken per moraine section.

## 3.3 Photogrammetric analysis

The archival aerial images as well as the drone images were used to generate DEMs by photogrammetric processing using the Structure from Motion with Multi-view stereo principle (SfM-MVS) in Agisoft Metashape Professional (https://www.agisoft.com/). The background and workflow of using SfM-MVS is explained in diverse publications (Aber et

al., 2019; Carrivick et al., 2016; Eltner et al., 2016; Smith et al., 2016; Westoby et al., 2012).

First, the drone images of each investigated moraine section were processed. After the calculation of the sparse cloud in Agisoft Metashape, the GCPs measured in the field (see section 3.2) were set directly on the images as markers and optimizations of the camera parameters were done several times. Subsequently, the dense cloud was calculated with high quality and orthophotos were generated.

The historical digitized (scanned) as well as the newer, digital aerial images, all covering the whole glacier forefields, were processed in the next step. As described in Stark (2020) and Altmann et al. (2020), especially the historical images were pre-processed by (i) resizing the images so that each set of images had the same size in order to be associated with one camera, (ii) masking of the black border of each image (not necessary for the newer, digital images) and (iii) contrast adjustments for some of the images. In the following, the images were processed with the same standard workflow as the digital, newer images. For

global referencing, first the GCPs selected for the drone image campaign were applied to the newest aerial images, since these have the smallest difference to the drone survey and better image quality. Once having processed this newest aerial image dataset, we used the orthophotos and DEMs to select GCPs well distributed within the total glacier forefield and on the margins which could also be identified in the historical aerial image datasets. The GCP coordinates were extracted from the already generated DEMs and applied to the historic aerial images.





## 3.4 Point Cloud Adjustment and Calculation of DEMs of Difference (DoD)

The further processing of the points clouds was done in SAGA LIS (Conrad et al., 2015; laserdata.at/software). After cropping the point clouds to the extent of the area of interest and removing outliers, the point densities of point cloud pairs selected for comparison were set to a similar value via 3D block thinning. This enables a better adjustment of the point clouds and reduces the data amount (Mayr et al., 2018; Stark et al., 2020). In a next step, preliminary DEMs and DEMs of difference (DoD) were employed to visually identify stable areas.

In the following, the points within these stable areas were used to optimise the co-registration of the point clouds using the tool "iterative closest point adjustment" (ICP) in SAGA LIS. Subsequently, the calculated transformation and rotation matrix for the stable areas was applied to the whole point cloud which should be adjusted. In some cases, after the first ICPs the mean value of the DoDs was near zero, but systematic errors could be found in different parts of the model due to the fact that the ICP algorithm cannot eliminate complex, non-linear errors (cf. Bakker and Lane, 2017). Therefore, the affected point clouds were split into parts with either positive or negative deviations, and ICP was done separately before merging the adjusted subsets to one single point cloud.

After adjustment, raster DEMs were generated from the point clouds, using the mean elevation of points for each raster cell. The resolution was 1 m or sometimes 1.5 m for the DEMs of the aerial images (2.5 m for the ALS data of South Tyrol) and 0.2 m for the drone-DEMs. Finally, DoDs were calculated by subtracting the two corresponding DEMs.

## 3.5 Error estimation

Despite accurate ICP adjustment, the calculated DoDs still have an uncertainty which has to be estimated when an erosion rate is calculated. The uncertainty assessment is based on stable areas within the DoDs and was conducted following Anderson (2019) who proposes the estimation of two random errors (one spatially autocorrelated and one uncorrelated) and one systematic error. A test was done with four different DoDs of this study (representative combinations of DEMs generated from aerial images, drone images, ALS and TLS) in order to quantify the different error types. The results show that the uncorrelated and correlated random error are very small (0.003 cm–1.7 cm) (cf. Betz-Nutz, 2021) and thus only the systematic error was used for the error estimation. For the calculation of the latter, different error metrics are possible and we decided for the standard deviation (precision) and the mean value (accuracy) of the DoDs (cf. James et al., 2019).

## 3.6 Investigation of geomorphic processes

We used the orthophotos, DEMs and DoDs to qualitatively and quantitatively investigate geomorphological processes occurring on the moraine slopes.

### 3.6.1    Interpretation of DoDs and orthophotos





The DoDs and orthophotos were interpreted regarding the surface changes that have occurred. Since surface changes are well
visible in DoDs, their spatial patterns give important hints e.g. on the type of geomorphic processes. The orthophotos aid the
interpretation of the patterns seen in the DoDs. Together, this enables an evaluation about what geomorphic processes are
affecting and forming the lateral moraines.

Moreover, the DoDs are helpful for the mapping of glacier extents and the detection of melting dead ice. The detection of
debris covered ice, still in contact with the glacier or already isolated, is often difficult when looking only at the surface at one
point of time, e.g. on an orthophoto. The use of DoDs can be very helpful for the detection of debris covered ice as they show
negative values for subsiding surfaces (Abermann et al., 2010; Schomacker, 2008). This enables the exclusion of areas affected
by debris covered ice from analyses of geomorphic processes and the computation of erosion rates.

Thus, DoDs were used for the glacier mapping supplementary to the orthophotos back to the 1950s. The glacier extents were
mapped in a way that ice on the margins which is covered by debris, but still in contact with the glacier as visible on DoDs, is
considered as part of the glacier (see example in Betz-Nutz, 2021, p.71, Fig. 12). For the mapping of earlier glacier extents,
old maps and descriptions in the literature were used (see section 3.1).

Regarding the detection of melting dead ice with the help of DoDs, there are some features which can help to differentiate it
from the melting glacier and sediment erosion (see also Fig. 2): First of all, it is isolated from the glacier mass. Furthermore,
melting dead ice is usually represented by relatively high values in the DoD compared to sediment erosion values. Moreover,
melting ice appears not punctually or linear, but in larger areas. Geomorphic processes such as landslides, in contrast, would
change the texture of the surface and retreated moraine ridges would be visible. Instead, melting ice below the surface inhibits
the formation of rills or other forms due to the constant change of the surface by the thawing. Another point is that a surface
depression caused by dead ice melt-out lacks corresponding depositional features in the downslope, while eroded material
would have been deposited somewhere below or a channel would have transported the sediment away. Thus, if a possible
sediment erosion can be excluded by the analysis of these features, the melting of debris covered ice is very probable.



Earth **Surface**
**Dynamics**
Discussions



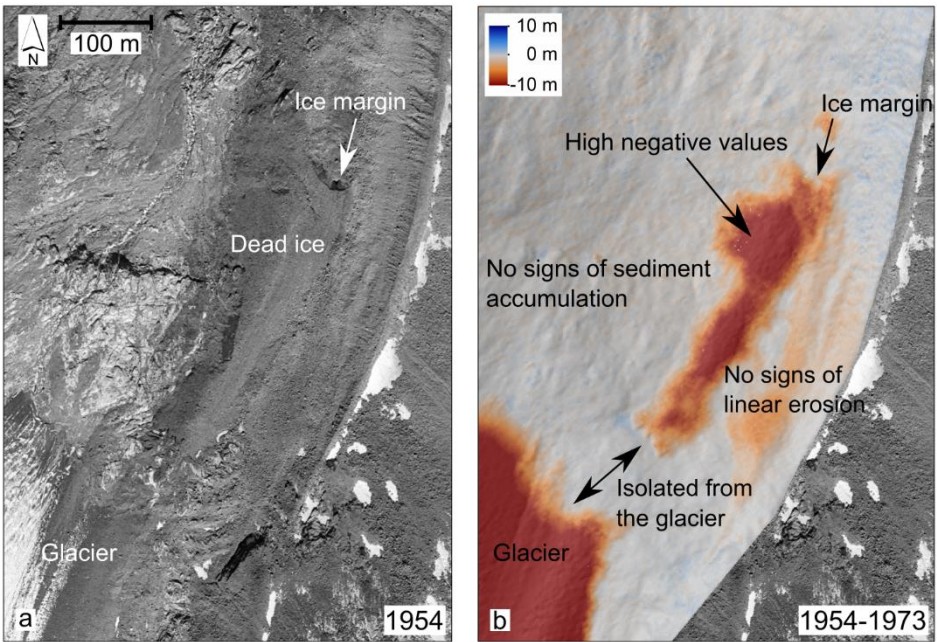

**Figure 2: Detection of debris covered dead ice using the example of the Alpeiner Ferner: (a) Orthophoto of 1954 and (b) DoD of 1954–1973 (Sources of aerial images: 1954 BEV, 1973 Office of the Tyrolean government)**

### 3.6.2 Morphometric analysis

Moreover, we compared profile lines of different DEMs and calculated the gully depth of all moraine sections with gullies as well as the headward retreat rate on these slopes.

For the calculation of the maximum depth of the gullies, first the most recent DEM was used to calculate the flow accumulation, contour lines and a hillshade. On the basis of these datasets, lines were digitized orthogonal to the gully thalweg from one gully ridge to the next, and in slope parallel distances of about 1 m. Then, the height differences between the ridge and the thalweg were measured and the profile with the biggest height difference was determined. Moreover, the slope gradient of the deepest point of the respective profile in the thalweg was determined. Its cosine was then used to correct the height difference in order to determine the incision perpendicular to the gully thalweg.

Nearly all moraine slopes with gullies show headward retreat. In order to gain insight into the rates of the headcut retreat, the lines connecting the gully headcuts were mapped on the oldest (1950s) and the newest (2018/2019) orthophotos. In a next step, the minimum and maximum Euclidian distances between the headcut lines on the old and new orthophotos were measured. To approximate the mean retreat rate, the area between the lines was divided by the width of the gullied moraine section.

### 3.6.3 Quantification of erosion rates



To enable a comparison of the morphodynamics on the lateral moraines, it was necessary to define comparable areas and time
periods for the different study areas and moraine sections. Due to different sizes of the moraine sections and different
proportions of the erosion-, transport- and deposition areas, the erosion rates were only quantified on the erosion areas on the
upper slopes of the selected moraine sections, representing the main areas of erosion.

For that purpose, on each moraine section the erosion area was mapped. The upper border of each erosion area was either the
moraine ridge or the headcut of the gullies present on the slope. The lateral borders were mostly equal to the moraine section
outline (see section 2). The lower border of the erosion areas was drawn at the transition from the erosion to the transport area,
which corresponds to the downslope end of the gully sidewalls on gullied slopes (see Fig. 7). For moraine sections which show
so little erosion that no erosion area can be clearly defined, the entire moraine section is considered as erosion area.

As slope areas including dead ice melting would falsify the calculation of the real sediment erosion, dead ice areas were
excluded from the mapped erosion areas. In cases where it was not possible to clearly delimit the dead ice-affected area from
the ice-free slope, no erosion area could be defined for these moraine sections.

Wherever possible and reasonable, the outlines of the erosion areas were used for several DoDs of the same moraine section,
as this ensures best comparability. However, for the case of extensive changes over time, the erosion areas had to be defined
independently for different DoDs in order to include areas formerly featuring dead ice/glacier areas or newly formed erosion
areas in younger DoDs.

Subsequently, all DoD cells with negative values within the erosion areas were summed up, multiplied by the cell size and
divided by the size of the erosion area and the number of years spanned by the respective DoD. Hence, the resulting erosion
rate represents a mean annual surface elevation change (in cm a$^{-1}$) within a certain period.

In order to improve the comparability and ease the interpretation of the temporal evolution of erosion rates, the DoDs were
allocated to three general time periods of similar length, based on the majority of the available image data sets:

- Time period 1: All DoDs from the 1950s to the 1970s
- Time period 2: All DoDs from the 1970s to the 2000s. DoDs ranging from the 1950s/1960s to the 2000s are
    allocated to period 1 and 2 (same erosion rate for both time periods)
- Time period 3: All DoDs from the 2000s to 2018/2019. The erosion values calculated on base of several DoDs
    within this time period were summed up and divided by the years covered by the entire period.

A list of all DoDs used for the calculation of erosion rates and their attribution to the time periods can be found in Table 3.
Note that some moraine sections of the DoDs covering the entire forefields are affected by dead ice in some time periods (see
Fig. 9) and for some moraine sections additional drone images were acquired (years in parentheses).

**Table 3: DoDs used for the calculation of the erosion rates and their attribution to the defined time periods 1–3**

| Study area | DoDs of time period 1 | DoDs of time period 2 | DoDs of time period 3 |
|---|---|---|---|
| **Vedr. d'Amola** | Glacier/Dead ice melting | 1983–2004 | 2004–2017, 2017–2018 |
| **Alpeiner Ferner** | 1954–1973 | 1973–2009 | 2009–2018, 2018–2019 |
| **Gepatschferner** | 1953–1971 | 1971–2014 | 2014–2015, 2015–2017 |



| Hohenferner | 1959–2006 | 1959–2006 | 2006–2016, 2016–2017, 2017–2018, (2018–2019) |
|---|---|---|---|
| Höllentalferner | 1960–2009 | 1960–2009 | 2009–2018, 2018–2019 |
| Krimmler Kees | Glacier/Dead ice melting | 1974–2006 | 2006–2017, 2017–2018 |
| Langtauferer Ferner | 1959–2006 | 1959–2006 | 2006–2016, 2016-2017, 2017–2018, (2018–2019) |
| Rofenkarferner | 1953–1971 | 1971–2009 | 2009–2017, 2017–2018 |
| Waxeggkees | 1954–1971 | 1971–2010 | 2010–2017, 2017–2018 |
| Weißseeferner | 1953–1971 | 1971–2014 | 2014–2015, 2015–2017 |

## 3.7 Analysis of possible influencing factors on morphodynamics

For the investigation of possible factors influencing the morphodynamics on lateral moraines, besides the detailed analysis of the moraine sections, we conducted also an analysis including the entire defined glacier forefields. This analysis is based on the negative raster cells of the DoDs and parameters derived from the DEMs.

For this analysis, the most recent DoD covering the entire respective glacier forefield was used, representing mainly the second analysed time period ranging from the 1960s or 1970s to the 2000s. The most recent DEMs of the entire glacier forefields, so

from the 2000s, formed the basis for the derivation of the slope angle. The resolution of all models is 1 m. All areas with bedrock and areas with glacier or dead ice were excluded from the analysis as well as river channels, forest areas and human infrastructure.

In order to analyse the relationship between the erosion and time since deglaciation, an interpolation of the mapped glacier extents was necessary to get a value for the time since deglaciation for each raster cell. For this purpose, we gridded the mapped

glacier outlines and calculated a distance grid showing for each pixel the Euclidian distance to the next pixel of a glacier outline. In a next step, we used two distance grids of subsequent glacier extents to calculate for each pixel the year of deglaciation using a linear interpolation of the respective dates of the glacier extents based on the distance ratio. After this was calculated for each pair of distance grids, all grids with the years of deglaciation were merged. Moreover, the years since deglaciation were approximated by the difference of the date of deglaciation of the respective raster cell and the mean year of

the time period spanned by the used DoD.

## 4 Results and discussion

### 4.1 Results of the error estimation

For the quality assessment of the different DoDs, an error estimation was conducted within stable areas, which revealed a wide range of errors depending on the type of dataset. Figure 3 shows boxplots with the distributions of the mean values and standard

deviations of the DoDs. Both parameters are highest for the DoDs based on aerial images. The absolute value of the mean ranges between 0.2 and 18.4 cm within this group and the standard deviation between 12.2 and 76.5 cm. The large differences



within this type of DoDs can be explained by the different quality of the aerial images and the resulting surface models. Within the group of DoDs generated from a DEM based on aerial images and a DEM based on drone images, the absolute value of the mean ranges from 0.03 to 4.8 cm and the standard deviation from 5.3 to 24.6 cm. This outlier of 24.6 cm can be explained

by an aerial image dataset of low quality within the Vedretta d'Amola area (2004). However, the variance is much lower within this group. An even lower variance is visible for the DoDs based on drone image datasets, where the absolute value of the mean is 1.8 cm at most and the standard deviation between 1.3 and 6.5 cm. The uncertainties determined for the aerial image datasets are comparable to the values reported by Stark et al. (2020) and Micheletti et al. (2015a) and this is also the case for the DoDs based on of drone images (e.g. Smith and Vericat, 2015).

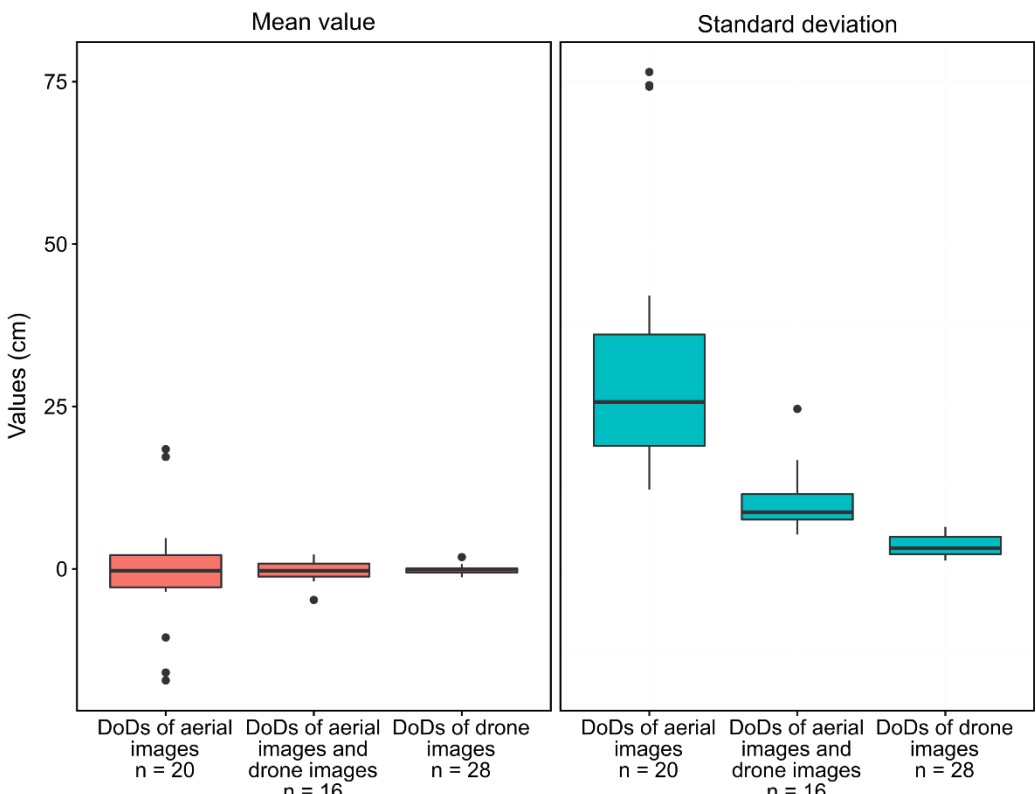

**Figure 3: Boxplots showing the distribution of the mean values and standard deviation for different types of DoDs: DoDs calculated on base of aerial images, DoDs calculated on base of aerial and drone images and DoDs out of two sets of drone images**

## 4.2 Occurrence of dead ice

Melting dead ice is widespread within the investigated glacier forefields and thus in the lateral moraines; it was found in all

but one glacier forefield, that of the Höllentalferner. Both the long time span and the low elevations in which dead ice melt-out was detected in some forefields are surprising. In the forefield of the Alpeiner Ferner, at moraine section APF2, dead ice is detectable from 1953 to 2019, i.e. for 65 years, at an altitude of about 2500–2600 m a.s.l. In the forefield of the Krimmler Kees, dead ice existed for several years (at least from 2004–2018) at an altitude of about 2000–2200 m a.s.l. This shows that



the lateral moraines are strongly affected by melting dead ice. In the following, we present two examples of different reactions
of lateral moraine slopes to melting dead ice.

The first one is a moraine slope in the forefield of the Hohenferner in the Martell valley, where dead ice was melting on the
middle and lower slope between 1959 and 2016 (Fig. 4a). When comparing the length profiles of 1959 and 2016 (Fig. 4b), a
depression by up to 7 m is visible in the middle and lower slope part, created due to the unoccupied space after ice melting.
Moreover, an increase of the slope gradient by about 10° is detectable in the area where dead ice melted. In total, the surface
subsidence leads to a withdrawal of support for the steep upper slope and thus contributes to the destabilization of the slope.
It can be supposed that the incision of gullies on the upper slope has been enhanced by the subsidence on the middle and lower
slope.

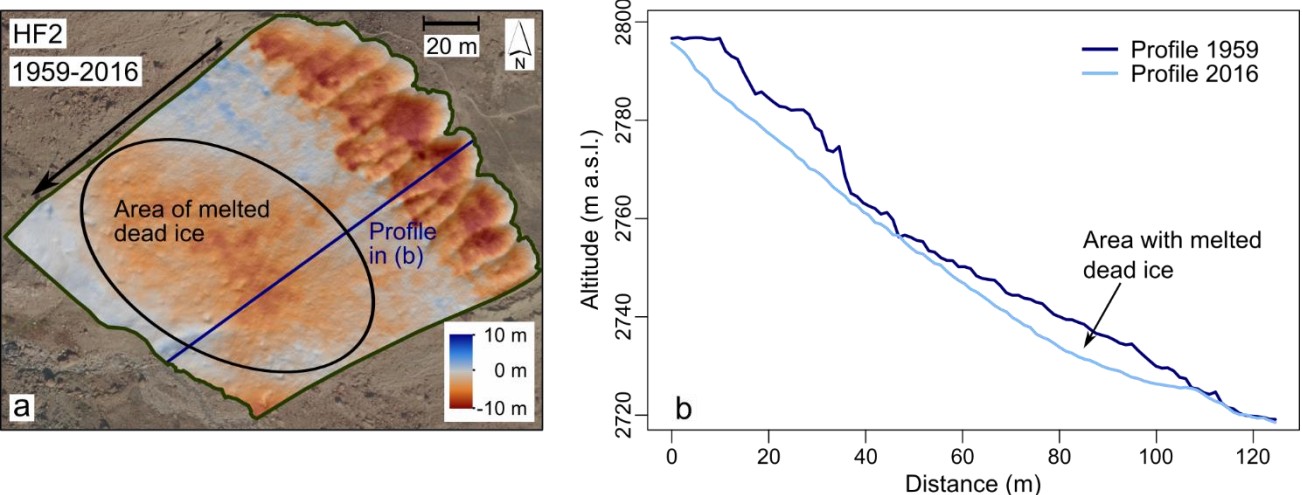

Figure 4: Reaction of lateral moraine slopes to dead ice melting on base of the example of moraine section HF2: (a) DoD of 1959-
2016 and (b) length profile through the DEMs of 1959 and 2016 (Sources of aerial images: 1959 IGMI, 2016 Hydrographic Office,
Agency for Civil Protection, Autonomous Province of Bolzano-South Tyrol)

The moraine section KK3 in the forefield of the Krimmler Kees shows a different reaction to melting dead ice on the middle
and lower slope. On this slope, no incision of gullies has occurred and generally hardly any linear erosion is detectable.
Possibly, dead ice reaching further up the slope than it is visible nowadays inhibited the formation of linear erosion forms.
Instead, the complete slope seems to slide down which is detectable from high negative values on the middle slope and the
slightly positive values of the DoD on the lower slope, probably compensated partly due to melting ice on the lower slope (Fig.
5a). On the lower slope part, the sliding is also clearly visible by some bigger rocks which slid down by 2–3.5 m. We infer
from the existence of vegetation (Fig. 5b) that geomorphic activity beyond the subsidence is low – otherwise the vegetation
would have been affected.
Looking at all investigated glacier forefields, the subsidence due to melting dead ice causes the largest height differences in
our DoDs (cf. Sailer et al., 2012), besides of glacier melting. A retarding influence on the stabilization of the moraines by the





melting ice and the induced geomorphic processes can be strongly supposed, as also assumed by Lukas (2011) and shown in Ravanel et al. (2018).

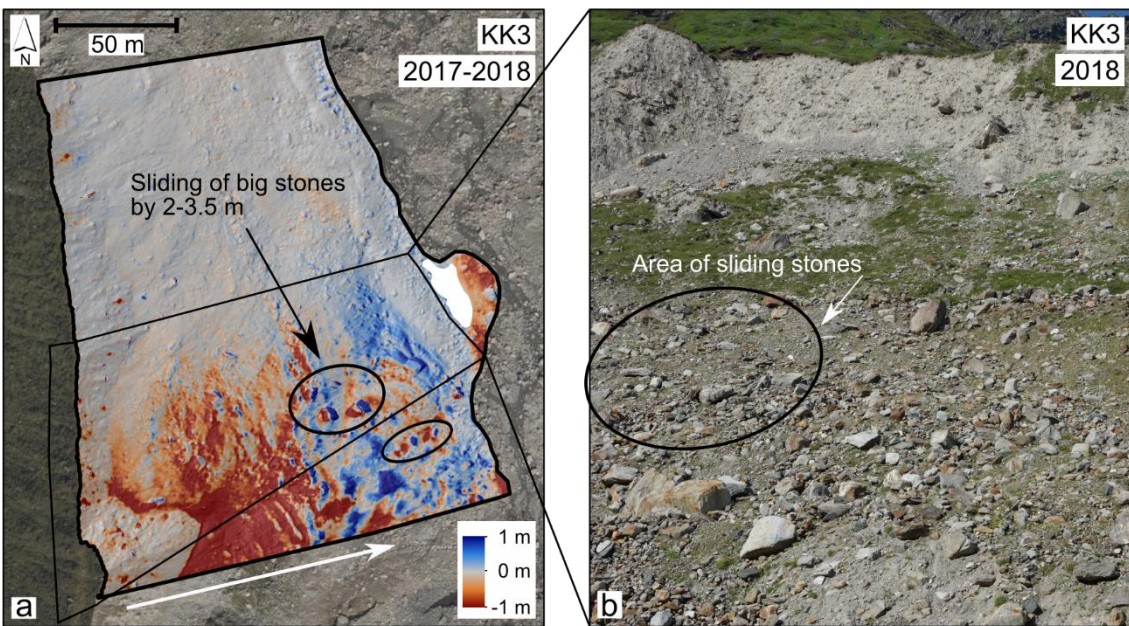

**Figure 5: Reaction of lateral moraine slope to dead ice melting on base of the example of moraine section KK3: (a) DoD of 2017–2018 and (b) image taken from the foot of the moraine slope towards the ridge on 01.08.2018, rough outline marked in (a)**

### 4.3 Geomorphic processes and landforms

Based on the interpretation of (i) the spatial pattern of negative and positive values of the DoDs and (ii) orthophotos, different types of geomorphic processes and landforms on the moraine slopes could be detected. In the following, three main types of 360 slopes with typical predominant geomorphic processes and landforms will be discussed.

#### 4.3.1 Landslides

In some of our study areas, landslides occurred within the study period beginning in the 1950s. One example in the forefield of the Langtauferer Ferner has been detected on the DoD of 1959–2006 and the orthophotos of 1959 and 2016 (Fig. 6b, c). Figure 6a shows the DoD of 1959–2016 (for reasons of better resolution). The landslide has a width of 145 m and the depth, 365 meaning the difference between the surfaces of 1959 and 2016, is up to 30 m in some areas. Ongoing erosion following the landslide could have caused further incision, so the original landslide may have been slightly smaller. The landslide displaced the moraine ridge backwards by up to 40 m (orthophotos in Fig. 6b and c). As the DoD shows, the areas below the landslide on the valley bottom were still affected by glacier and dead ice melting within this period. This leads to the assumption that the landslide could have been a reaction to changing erosional base levels. It is conspicuous that other glacier forefields that 370 feature big landslides, such as the Gepatschferner with a landslide of a similar size, were also substantially ice covered on the valley bottom at that time. Other glacier forefields that were already deglaciated at the starting point of our investigations,

however, showed no landslides during our observation period and additionally the earliest orthophotos or DEMs do not show indications of displacements of moraine ridges prior to ca. 1950 that could correspond to landslides. In the forefields of the Langtauferer Ferner and the Gepatschferner landslides seem to be the dominant processes during deglaciation regarding the

erosion volume, as it was also observed in the study of Cody et al. (2020) in New Zealand.

Gravitational deformation of moraines leading to large gaps in the moraine crest during glacier recession are also reported by Hugenholtz et al. (2008) in the Canadian Rocky Mountains. Moreover, Blair (1994) describes slope failure of lateral moraines and valley walls in New Zealand as a consequence of accelerated ablation rates of the glacier.

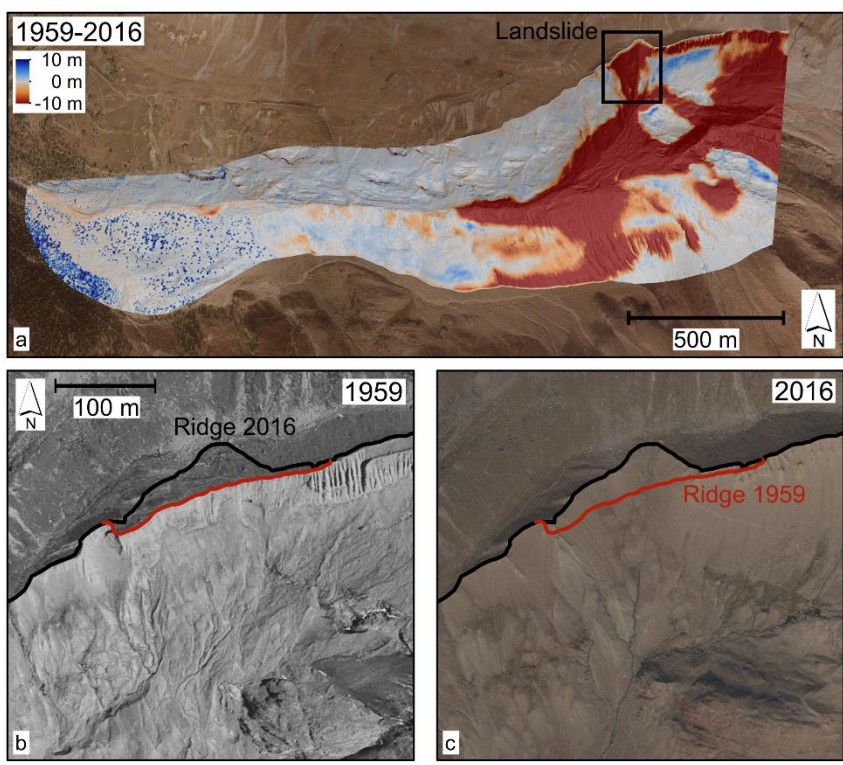

**Figure 6: Landslide on the lateral moraine of Langtauferer Ferner: (a) DoD 1959–2016 of the entire glacier forefield, (b) aerial image of 1959 and (c) aerial image of 2016 (Sources of aerial images: 1959 IGMI, 2016 Hydrographic Office, Agency for Civil Protection, Autonomous Province of Bolzano-South Tyrol)**

### 4.3.2 Slopes with deeply incised gullies

Many of the investigated glacier forefields exhibit lateral moraine sections with heavily gullied upper slopes. All these moraine
slopes were affected either by a melting glacier or melting dead ice on their middle and lower slope within our observation period, except of two moraine sections which deglaciated already soon after 1900. At several moraine sections, the formation of the heavily incised gullies could be observed beginning with our first DoDs from the 1950s to the 1970s (APF3, GPF1, GPF2, HF2, KK2, LTF2, LTF3, WSF2, for the location of the moraine sections see Fig. 1). On some sections, however, gullies were already present in the 1950s (beginning of observation period) and they do not seem to have experienced further gully





incision since then (HTF1, WSF1, WEK2, WEK3). For the gully formation since the 1950s, the moraine section APF3 in the
forefield of the Alpeiner Ferner serves as an example (Fig. 7).

The orthophoto of 1954 (Fig. 7e) and also a cross profile through the DEM of 1954 (Fig. 7i) show that at that time only small
rills were present at the upper slope, whereas the lower and middle slope were covered by the glacier in 1954. As the DoD
1954–1973 shows (Fig. 7a), erosion took place in these rills, presumably especially linear fluvial erosion. Already the
orthophoto of 1973 (Fig. 7f) shows an enlargement of the rills that became gullies over time. On the DoD 1973–2009 (Fig.
7b), it is also visible that not only linear erosion incising the rills, but also a widening due to erosion on the gully sidewalls
occurred. It can be assumed that besides fluvial erosion also small-scale gravitational processes such as sheet erosion by snow
gliding and denudation lead to the gully widening. The DoDs of 2009–2018 and 2018–2019 (Fig. 7c and d) show that debris
flows have become an important process that causes further gully incision and widening. The slope has steepened from 1954
to 2018 from 36.9° to 43.5° and the glacier has nearly completely melted in the meantime (see also comparison of the length
profiles in Fig. 7j). In 2019, a gully depth of about 6 m could be measured (see cross profile in Fig. 7i). The accumulated debris
flow material is visible on the lower slope part, right before reaching the channel, indicating poor hillslope-channel coupling
in this place. The erosion on the gully heads, mainly happening after 1973, leads to the headward retreat of the erosion area
(marked in purple in Fig. 7e and h). This retreat amounts to between 4 and 40 m and is easily possible on this moraine slope
as it has no sharp ridge; the slope continues upward into a ground moraine of an earlier glacier flowing eastward. It can be
assumed that this favours the ongoing intense erosion and prohibits stabilization. An influence of the dead ice, which is
detectable on the DoDs 2009–2018 and 2018–2019, on the geomorphic activity is not clear in this case, as above the dead ice
area there is also a bedrock-outcrop visible since 2009 (area magenta in Fig. 7g and h) which contributes to stabilization.

These observed processes and the gully development on the moraine slopes in this study are consistent with the observations
of e.g. Dusik et al. (2019), Neugirg (2016) and Wetzel (1992) on similar slopes: Fluvial erosion starts with the formation of
rills; once the rills are bigger, loose material from the sides and headcuts is deposited in the rills e.g. by snow gliding and
finally small debris flows evolve from this loose material, supposedly mainly during heavy rainfall in summer, and contribute
to a widening of the rills to gullies. It seems that these small debris flows are the dominant processes of sediment transfer on
slopes with deeply incised and widened gullies in our study, as reported before (cf. Ballantyne, 2002b; Cody et al., 2020; Curry
et al., 2006; Dusik, 2020; Wetzel, 1992).





**Figure 7: Development of deeply incised gullies with the example of moraine section APF3 in the Alpeiner Ferner forefield: (a) – (d): DoDs of moraine section APF3 for 1954–1973, 1973–2009, 2009–2018 and 2018–2019 with the erosion area (yellow) and dead ice (cyan), (e) – (h): orthophotos of APF3 for 1954, 1973, 2009 and 2019 with length (blue) and cross profiles (red/orange), (i) Cross profiles through APF3 for 1954 and 2019 and (j) Length profiles along APF3 for 1954 and 2019 (Sources of aerial images: 1954 BEV, 1973/2009 Office of the Tyrolean government)**

Table 4 shows an overview of all gullied moraine sections with respect to (i) the headcut retreat of the gullies (minimum, maximum and mean for each moraine section) and the (ii) gully width, maximum depth and gradient of the gully thalweg of the deepest gully. A maximum headcut retreat of 85 m could be measured on a moraine section in the Langtauferer Ferner forefield (LTF2), followed by a moraine section in the Gepatscherferner forefield with a maximum retreat of 68 m (GPF1). The latter slope also stands out regarding the maximum gully depth: A very big gully on this moraine slope has a maximum depth of 17.5 m. This is also due to the relatively low slope gradient of 32° and the high gully width of 56.7 m. The gullies on the other moraine sections show depths of maximum 6.4 m (LTF2) down to 2.1 m (WSF1).

A collapse or widening of the gullies until a levelling of the slope occurs, as described in Curry et al. (2006) based on the comparison of moraine sections with different age since deglaciation, cannot be reproduced with the data in this study. Only some gullies show less activity over time and no incision any more (WSF1, WEK2, WEK3, HTF1), however, they do not



collapse and the slope is not levelled. As already compared by Betz et al. (2019), where the moraine sections LTF2, LTF3 and HF2 were analysed similarly as by Curry et al. (2006), these moraine sections do not show the progression of gully development with increasing age as described by Curry et al. (2006). Neither is a maximum gully incision reached 55 years

after deglaciation nor do the gullies widen only 80–140 years after deglaciation. Also Dusik et al. (2019) could not reproduce the gully development assumed in Curry et al. (2006).

**Table 4: Headcut retreat minima, maxima and mean values for each gullied moraine section and for the most incised gully on the respective moraine section maximum gully depth, corresponding gully width and corresponding slope gradient at gully bottom**

| Moraine section | Min. headcut retreat (m) | Max. headcut retreat (m) | Mean headcut retreat (m) | Max. gully depth (m) | Corresponding gully width (m) | Corresponding slope gradient (°) at gully bottom |
|---|---|---|---|---|---|---|
| APF3 | 4 | 40 | 21.4 | 6.0 | 16.2 | 37 |
| GPF1 | 9 | 68 | 33.2 | 17.5 | 56.7 | 32 |
| GPF2 | 7 | 22 | 13.2 | 4.7 | 16.8 | 42 |
| HF2 | 2 | 13 | 8.5 | 5.4 | 9.8 | 37 |
| KK2 | 6 | 13 | 8.6 | 4.6 | 9.8 | 52 |
| LTF2 | 8 | 85 | 43.8 | 6.4 | 17.9 | 43 |
| LTF3 | 2 | 17 | 8.8 | 3.8 | 5.8 | 48 |
| WEK2 | 1 | 5 | 3.5 | 3.3 | 10.8 | 41 |
| WEK3 | 7 | 17 | 13.3 | 2.6 | 12.5 | 48 |
| WSF2 | 0 | 4 | 2.1 | 3.6 | 14.5 | 42 |

### 4.3.3 Slopes with shallow incised gullies or no incision

Besides the moraine sections with deeply incised gullies as the one explained in section 4.3.2, there are lateral moraine sections which show only very shallow incised gullies, rills or even no incisions at all, but denudation, sheet erosion or no erosion at all. One lateral moraine section in the Alpeiner Ferner forefield, APF2, is an example showing only shallow incised rills (Fig. 8). Although there is dead ice melting on the middle slope, the upper slope (41° slope gradient) does not show heavy gullying.

Instead, mostly fluvial processes and small slides are occurring, maybe also small debris flows of short length, as it can be seen especially on the DoD 2018–2019 in Fig. 8b. It can be assumed that the melting dead ice, which also reached up to the upper slope until recently as suggested by the DoD 2009–2018 in Fig. 8a, inhibited the incision of linear forms due to the constant, areal melting of the ice. Without gully incision and headcut erosion, also a headcut retreat of the moraine ridge is missing here. A stabilization of the slope, however, has not yet occurred as the already existing grass vegetation is destroyed

again by the geomorphic processes (images in Fig. 8c and d).

Lateral moraine slopes with such kind of geomorphic processes, i.e. mainly fluvial processes and small slides, can be found in almost all investigated glacier forefields. However, mostly they do not show melting dead ice any more. Some of these moraine slopes show ongoing activity, but some also a decrease in activity and signs of stabilization such as vegetation growth. The development of the corresponding erosion rates is presented in the following section.



**Figure 8: Slope with little incised gullies or rills illustrated for moraine section APF2 in the Alpeiner Ferner forefield: (a): DoD 2009–2018 (Source of aerial images 2009: Office of the Tyrolean government), (b) DoD 2018–2019 with the erosion area (blue), (c) – (d): Drone images of the slope of 01.08.2019**

## 4.4 Erosion rates

The temporal development of the morphodynamics and thus the erosion rates differ between the investigated lateral moraine sections. Decreases of erosion rates as well as stagnations and increases in activity could be detected. Figure 9 compares the erosion rates of the 20 moraine sections for which erosion rates could be calculated at least for two of the three time periods (1950s–1970s, 1970s–2000s, 2000s–2018).





The highest erosion rates could be found for the moraine slopes with deeply incised gullies as described in section 4.3.2 (names
marked in bold in Fig. 9). Within this group, the highest erosion rate was detected on moraine section GPF1 with nearly
43 cm yr$^{-1}$ in the first time period. At this time the glacier (Gepatschferner) and dead ice melted on the slope foot and partly
also on the middle slope part. Another moraine section in the Gepatschferner forefield (GPF2) also shows high rates with
14 cm yr$^{-1}$ in this first observed time period. Also, two moraine sections in the Langtauferer Ferner forefield (LTF2, LTF3)
have an erosion rate of 14–15 cm yr$^{-1}$ in the period from 1959–2006, where especially LTF2 was heavily influenced by the
melting glacier. This means that the highest erosion rates can be detected in the phase of strong glacier melting on the slope
foots being accompanied by initial gully erosion and on some slopes also by landslides (see section 4.3.1). After this initial
gully formation and glacier melting, the erosion rate decreases to lower than 10 cm yr$^{-1}$, but stays on a high level with more
than 4 cm yr$^{-1}$ erosion even decades after deglaciation in time period three. The gullied moraine section HF2 shows a slight
increase or at least stagnation over many decades (8.0 cm yr$^{-1}$ 1959-2006, 8.3 cm yr$^{-1}$ 2006–2018), while APF3 even exhibits
a clear increase of the erosion rate from period one to three (10 cm yr$^{-1}$ 2009–2019). It deglaciated completely only recently,
in 2012, and is still influenced by melting dead ice which is likely to influence the activity. Only on three gullied slopes with
less deeply incised gullies the erosion rates declined below 4 cm yr$^{-1}$ with rates between 2.2 and 3.7 cm yr$^{-1}$ in the third period
(WSF2, WEK2, WEK3). These slopes already deglaciated in the 1930s and the gullies were well developed in 1953, so unlike
on other slopes, the gullies have stabilized over the decades. Overall, most of the gullied slopes show a decreasing, however
ongoing high activity even after decades, only few seem to stabilize after many decades.

The erosion rates of the group of gullied slopes is comparable with erosion rates reported in literature, except for HTF1, which
shows only very little activity: For the erosion in gullies on lateral moraines, studies in Norway determined 1.9–16.9 cm yr$^{-1}$
(Ballantyne and Benn, 1994; Curry, 1999), a study in Switzerland 4.9–15.1 cm yr$^{-1}$ (Curry et al., 2006) and a study in Austria
2.9–3.4 cm yr$^{-1}$ (Dusik, 2020). These lateral moraines for which erosion rates are reported had been deglaciated for 43–79
years at the time of observation, which is comparable with our investigated moraine sections. However, it must be kept in mind
that the methods to calculate the erosion rates differ between the studies. Whereas e.g. in Curry et al. (2006) and Curry (1999)
the erosion rates were deduced from the volume of the existing gullies, assuming that this represents the eroded volume, in
Dusik (2020) and in this study the erosion rates were calculated based on the height differences detectable in the DoDs.

The slopes with shallow gullies, rills or no gullies show lower erosion rates and more signs of stabilization can be found. In
the first observed time period, erosion rates of up to 7 cm yr$^{-1}$ (KK1) occurred on these slopes, however, on most of the slopes
the rate was below 5 cm yr$^{-1}$ in all time periods. In the third period, the erosion rate decreased to around 1–3 cm yr$^{-1}$ on the
moraine sections KK1, LTF1, RKF3, WEK1, WSF1 and RKF2. The highest activity in the third period was detected on AM1
with 4.8 cm yr$^{-1}$, whereas HF1, which is characterized by sheet erosion by snow, shows a constant erosion rate of 2.1–
2.5 cm yr$^{-1}$. Reasons for the overall lower activity of these slopes can be seen in the longer time since deglaciation and lower
slope angles due to a location closer to the LIA maximum glacier extent (see sections 4.5.1 and 4.5.2). The rates are comparable
with the ones in Dusik (2020) on lateral moraines at the Weißseeferner (Austria) without gully formation with 0.8–1.7 cm yr$^{-1}$
(Dusik, 2020).



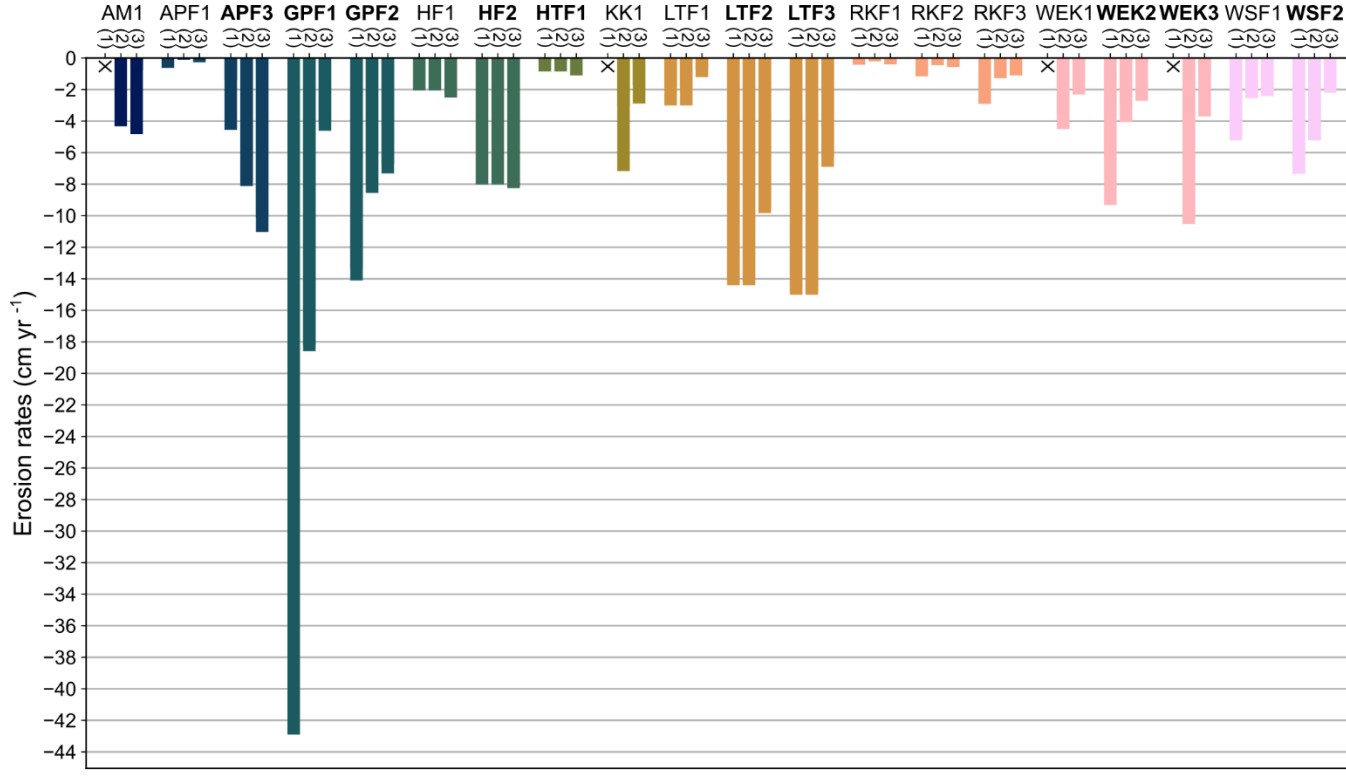

**Figure 9: Erosion rates for 20 investigated moraine upper slopes for the three time periods 1950s–1970s (1), 1970s–2000s (2) and 2000s–2018 (3) (see Table 3 for specific DoDs). Moraine sections written in bold represent upper slopes with deeply incised gullies (colour map batlow10 from Crameri et al., 2020)**

## 4.5 Influencing factors on morphodynamics

### 4.5.1 Time since deglaciation

As the time since complete deglaciation and dead ice melting on the moraines determines how much time has passed for the adjustment of lateral moraines and as well for landform evolution on the hillslopes, this parameter is important for the analysis of the morphodynamics.

In Fig. 10, the timing of complete deglaciation is plotted for all investigated moraine sections in a chronological order. Here, the end of dead ice melting is not considered as this is not detectable until the 1950s, the time of the first DEMs. Besides the

time of deglaciation, the erosion rates of the respective upper slopes of the moraine sections are indicated in this plot, blue colours showing erosion rates below 2 cm yr$^{-1}$, orange ones representing 2–4 cm yr$^{-1}$ and red text indicating moraine sections with more than 4 cm yr$^{-1}$ (moraine sections in black with undetermined erosion rates due to dead ice). It is conspicuous that the moraine sections with low erosion rates (below 2 cm yr$^{-1}$) deglaciated already before 1950, with one exception (RKF3). The moraine slopes with the highest erosion rates (red), however, deglaciated after 1950, with one exception (LTF3). However,



not all recently deglaciated moraine slopes have a high geomorphic activity and vice versa. For example, the moraine sections
       APF2, RKF3 and KK3 deglaciated quite late and two of them are still affected by dead ice melting, but on all these slopes no
       gullies have developed and the activity is rather low (observation based on orthophotos, where rates could not be calculated).

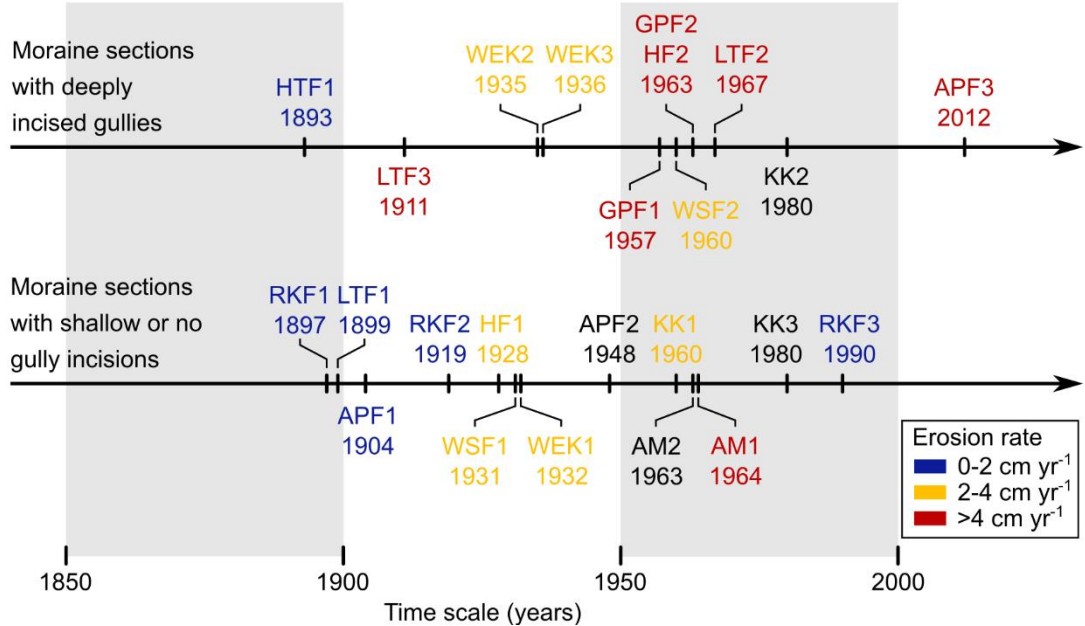

**Figure 10: Timeline with the years of deglaciation of the moraine sections, individually for slopes with deeply incised gullies and**
**slopes with only little or no gully incisions. Colouring of the moraine sections regarding the mean erosion rates of the DoDs in time**
       **period 3 (see Table 3 for specific DoDs; erosion rate of 0–2 cm yr⁻¹ in blue, 2–4 cm yr⁻¹ in orange and >4 cm yr⁻¹ red)**

       Besides this analysis of the investigated moraine sections, an analysis based on all grid cells within the 10 glacier forefields
       was conducted (see section 3.7). Figure 11 shows the sum of the annual erosion volume categorized by 20-year groups of the
       time difference between deglaciation and the mean year of the respective DoDs, divided by the entire area that deglaciated in
the respective time period. It is conspicuous that grid cells which deglaciated between the mean year of the respective DoD
       and up to 20 years before (mainly deglaciated between 1970 and 1990 for the DoDs used in this analysis) show comparatively
       low erosion. Going further back, the erosion rates increase, which means that a higher activity is detectable on older cells with
       respect to the time of their deglaciation. The erosion rate is then similar for a number of 20-year classes, but cells deglaciated
       for the longest time again show less activity. However, the rates stay higher than on the youngest grid cells. Most of the DoD
peaks in the respective glacier forefields can be explained by the high activity of certain moraine sections within the glacier
       forefield. For example, the high erosion values at the Alpeiner Ferner forefield between 20 and 80 years since deglaciation
       (orange bars) are largely attributable to the highly active moraine section APF3.



Earth **Surface**
**Dynamics**
Discussions

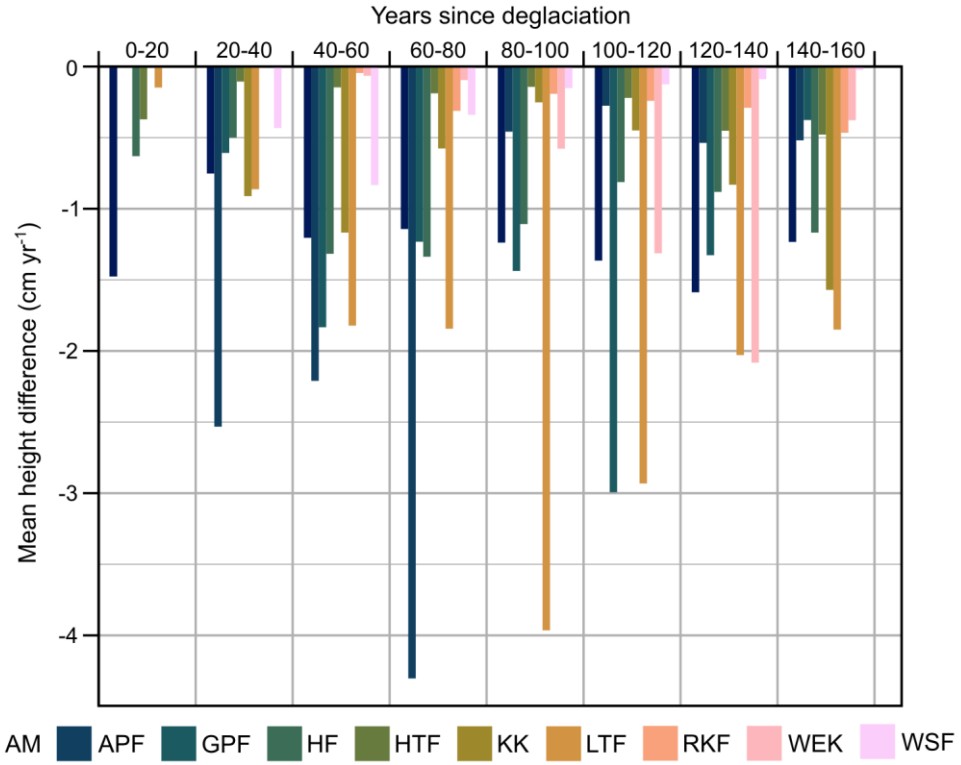

**Figure 11: Mean annual height difference (based on the DoDs of the study areas covering time period 2, see Table 3 for specific DoDs) for different 20-year groups since deglaciation (relating to the mean year of the used DoD; colour map batlow10 from Crameri et al., 2020)**

By comparing the results of the analysis for the investigated moraine sections (Fig. 10) and the grid cells in the glacier forefields (Fig. 11), it can be seen that in the former one a correlation of the erosion and time since deglaciation is detectable, whereas in the latter hardly any relationship can be observed. The missing correlation in the grid cell-based analysis can be explained by the simultaneous early deglaciation of the often highly active upper slopes and areas closer to the LIA glacier maximum which are, however, in general affected by only very little geomorphic activity. This shows that there is not a simple decrease of morphodynamics with increasing time since deglaciation, which indicates that the reason for higher erosion rates on moraine sections that deglaciated later (moraine sections marked red in Fig. 10) is likely not given alone by the shorter time since deglaciation.

### 4.5.2  Slope angle

**Correlation of the slope angle and geomorphic activity**

The analysis of the slope angle and its influence on the geomorphic activity clearly shows that there is a correlation. In Fig. 12, classes of the slope angle (10°-classes) are plotted against the mean annual height difference of the DoD (volume difference per area covered by the raster cells of the respective slope angle class). It is visible that from a slope angle of 20–30° on, the erosion rate increases until an angle of 60–70°. The highest erosion rates for the classes with the highest slope angle can be



attributed to very high erosion on very steep, but small areas. Though, if the absolute sum of the erosion volumes is observed, i.e. without a standardisation of the erosion volume on the area covered by the raster cells of the respective slope angle class (not shown here), most of the erosion can be detected on moraine slopes with slope angles of 30–40° (38 %) and 40–50° (30 %). In any case, only very little erosion takes place on moraine slopes with less than 30°.

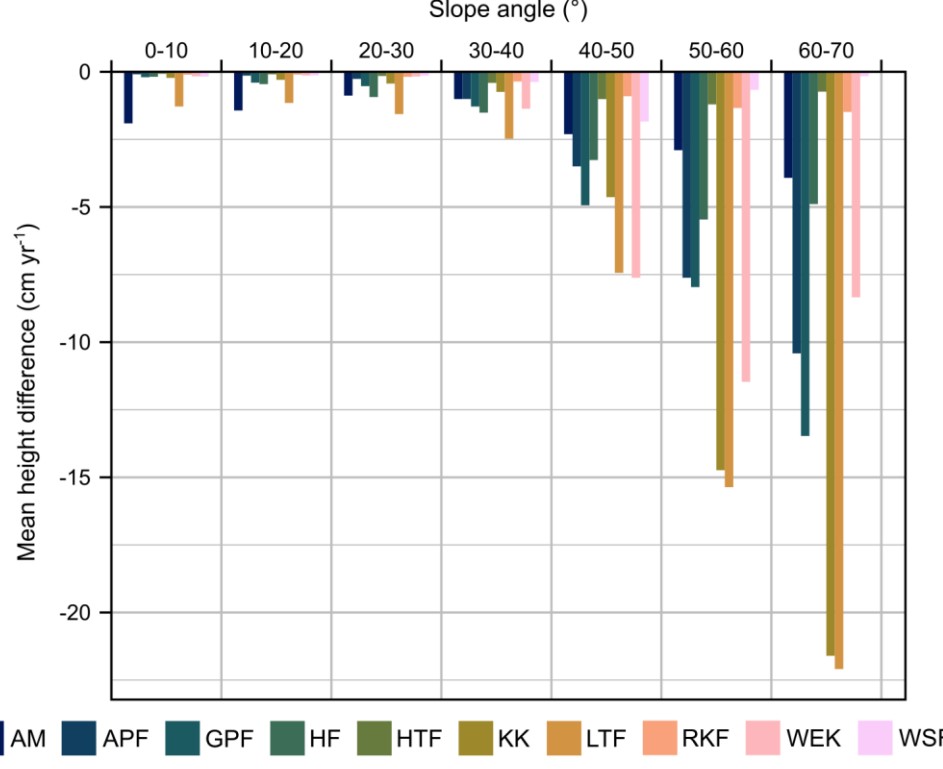

555

Figure 12: Mean annual height difference (based on the DoDs of the study areas covering time period 2, see Table 3 for specific DoDs) depending on the slope angle (colour map batlow10 from Crameri et al., 2020)

**Dependence of the slope angle on the distance from the LIA-maximum**

Since the analysis above revealed that the erosion rate depends on the slope angle, we also analysed where within the glacier
560 forefields particularly steep lateral moraine slopes can be found. For the Waxeggkees forefield, several cross profiles through the forefield show a decrease in slope angle with increasing distance from the LIA-maximum location (see Fig. 13a and b). The same trend can be observed when looking at the mean slope angle at different distances from the LIA maximum in all investigated glacier forefields (Fig. 13c). So, the mean slope angle increases in all analysed glacier forefields with increasing percentage of the distance from the LIA glacier maximum within the study areas, even though with different intensity and not
565 always in a linear form. These variations can be explained e.g. by bedrock areas which appear within the moraine tracts and have been excluded from the analysis. In any case, the slope angles are lower near the LIA maximum (0–10 %) than in further distance from the maximum and this means that the potential of higher erosion rates also increases with increasing distance from the LIA glacier maximum.





**Figure 13: Slope angle depending on the distance from the LIA-maximum: (a) Oblique view of the Waxeggkees forefield with cross profiles in different distances to the LIA maximum (source of aerial image 2010: Office of the Tyrolean government), shown in (b) against the altitude a.s.l. and (c) mean slopes angles in dependence on the distance from the LIA maximum for all glacier forefields (100 % refers to the maximum distance from the LIA maximum within the respective study area, mean slope angle for each distance-class based on the newest DEM of the study areas; colour map batlow10 from Crameri et al., 2020)**

**Change of the slope angle over time**

The findings shown above raise the question whether the decrease of the slope angle with increasing distance to the recent glacier tongue is a consequence of the longer time since deglaciation, since more time has already passed for adjustment, or of the genesis of the respective lateral moraines. For that purpose, the change of the slope angle over time was analysed for 20 upper slopes of the investigated moraine sections by a comparison of the slope angle in about 1950 (first available DEM) and 2018. In Fig. 14, the slope angle of ca. 1950 (y-axis) is plotted against the difference of the slope angle between ca. 1950 and





2018 (in °, x-axis). Moreover, each moraine section is coloured by the magnitude of the erosion rate within the time period of the 1970s to the 2000s (red for >4 cm yr$^{-1}$, orange for 2–4 cm yr$^{-1}$, blue for 0–2 cm yr$^{-1}$). The plot shows that the moraine sections with the highest erosion rates (red) are characterized by an increase of the slope angle on the upper slopes, which is up to 6.6°, with one exception (GPF1). All but one of the moraine slopes with moderate erosion rates (orange) show a decrease

of the erosion rates, mostly of 0.2–2°, but the slope angle on one slope decreased even by 3.2° (HF1). The moraine slopes with low activity (blue) show comparably little change of the slope angle due to very little sediment transfer. Overall, a clear decrease of the slope angles, following the idea of the paraglacial adjustment, is not detectable on the investigated upper slopes, especially not for the highly active moraine slopes, whose already steep upper slopes of 35°–45° steepened even more between ca. 1950 and 2018.

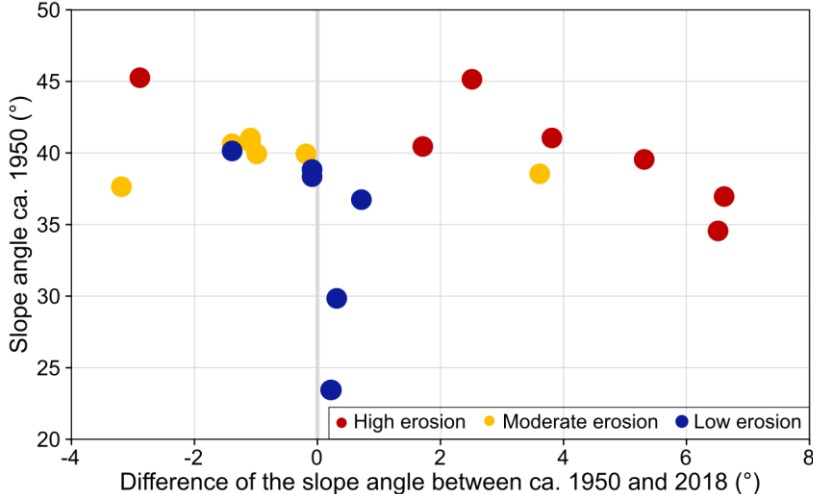


**Figure 14: Change of the slope angle between ca. 1950 (1953/1954/1959/1960, depending on the DEM) and 2018 for the upper slopes of the investigated moraine sections. Colouring of the dots represents the mean erosion rates of the DoDs in time period 3 (see Table 3 for specific DoDs; erosion rate of 0–2 cm yr$^{-1}$ in blue, 2–4 cm yr$^{-1}$ in orange and >4 cm yr$^{-1}$ red)**

The result that the slope angles of lateral moraines do not decrease substantially over decades, but presumably differ

significantly already from the beginning of deglaciation, is supported by old photographs from about 1900 showing the glacier forefields at a time, when the glaciers still covered substantial parts of the LIA extent. Already at that time, the lateral moraines were less steep near the LIA maximum, but steep slopes deglaciated further upwards, as visible e.g. for the Waxeggkees forefield in Fig. 15. Thus, the correlation between the slope angle and the distance from the LIA maximum, as shown above, is already visible at that time. This can be explained by the lower glacier ice thickness near the LIA maximum, which inhibits

the forming of lateral moraines with high slope angles, unless certain conditions of the relief, such as larger bedrock areas, lead to a congestion of the ice masses also near the LIA maximum (observable e.g. in the Langtauferer forefield).





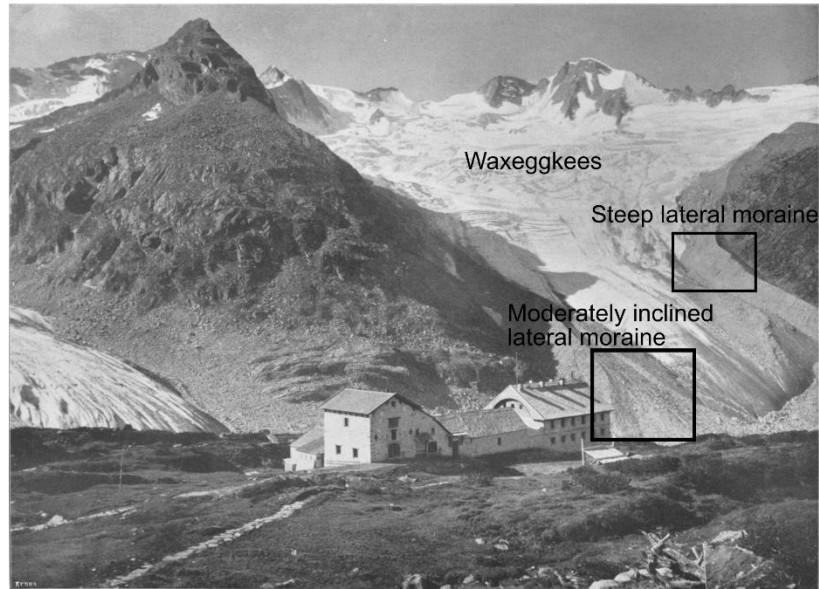

**Figure 15: Photograph of the Waxeggkees showing lateral moraines with decreasing slope angle in the direction of the LIA maximum extent within the valley (Würthle & Sohn, Salzburg, in: PLATZ & ROTHPLETZ 1903; by the courtesy of the Kulturarchiv Oberengadin)**

## 4.6 Implications of the results for the paraglacial adjustment

The majority of the study areas shows a decline of the erosion rates over time from the 1950s to the 2000s. However, there are moraine slopes with an increase or a stagnation of the erosion rates (see Table 5). Moreover, the erosion rates stay on a high level on some of the moraine sections. The fact that gullied slopes, which have already been deglaciated for 66 years, show still increasing erosion rates and those, which have been deglaciated for 120 years still show erosion rates of about 7 cm yr$^{-1}$ (LTF3), illustrates how long a stabilization on very steep and highly active slopes can take. The time since deglaciation for these gullied, very steep and active slopes is not long enough yet to detect a stabilization. Since there are no slopes with deeply incised gullies in the study areas that feature even longer times since deglaciation, the future development of the steepest moraine sections cannot be predicted based on the available data.

Furthermore, a correlation between the development of the erosion rates and the time since deglaciation is unclear, as shown in Fig. 11 and Table 5 (comparison of change of erosion rate and years since deglaciation). For example, the most recently deglaciated moraine sections (59–66 years) developed diversely. Thus, the rapid paraglacial adjustment of gullied slopes after some decades, as postulated by Curry et al. (2006), Ballantyne (2002a) and Eichel (2018) could not be generally proven, as several moraine sections did not show an increasing stabilization with ongoing time since deglaciation (e.g. HF1, LTF3, APF3). However, a tendency to paraglacial adjustment over longer time periods than reported in the mentioned publications is indicated by our data.

It is probable that the differences between the results of this study and previous studies are due to the fact that the latter used space-for-time substitution instead of an observation over longer time periods. Using space-for-time substitution for the study





areas of this work would have generated similar results regarding the erosion rates, because the slopes closer to the LIA
maximum (older in respective to the time since deglaciation) show smaller erosion rates than the ones at larger distance (see
comparison of activity today and years since deglaciation in Table 5). It should be noted though, that with the use of space-
for-time substitution different initial conditions of the slopes at the time of deglaciation are not reflected, as also described in
Wojcik et al. (2021) for the investigation of vegetation succession. However, these can have a substantial influence on the
morphodynamics. As our results showed, especially the slope angle is a crucial factor determining the morphodynamics on a
moraine slope. When comparing moraine sections in different distances to the LIA maximum, as it is done when using space-
for-time substitution, slopes are compared which did probably not have comparable slope angles at the time of deglaciation
due to the glacier ice mass distribution. Thus, it is clear that less inclined slopes with little distance to the LIA maximum also
exhibit lower erosion rates and show earlier signs of stabilization and vegetation succession. However, this is not necessarily
or not solely due to a longer time since deglaciation.

These results show that high slope angles are a necessary precondition for high morphodynamics. Further influence factors
can determine if a steep lateral moraine indeed shows high erosion rates and e.g. heavy gullying. The factors altitude a.s.l.,
slope undercutting by channels, climate and geology were investigated in Betz-Nutz (2021) and did not show a clear influence
on the morphodynamics on the lateral moraines. Factors such as the local substrate, the potential for headward retreat and also
non-local factors as the size of the catchment should be investigated further.

**Table 5: Overview over the slope angle, erosion rate, recent activity and years since deglaciation for all investigated moraine sections.**
**Change of the slope angle is indicated, if it changed more than 1° (median) between ca. 1950 and 2018, a change of the erosion rate,**
**if it exceeds 1.8 cm between the first and third period, and the recent activity is based on time period 3 (++ for >4 cm yr⁻¹, + for 2–**
**4 cm yr⁻¹ and 0 for 0–2 cm yr⁻¹)**

| | AM1 | APF3 | KK1 | WSF2 | HF2 | GPF2 | LTF2 | GPF1 | WEK2 | WSF1 | WEK3 | APF1 | RKF3 | LTF1 | LTF3 | WEK1 | RKF1 | RKF2 | HF1 | HTF1 |
|---|---|---|---|---|---|---|---|---|---|---|---|---|---|---|---|---|---|---|---|---|
| Change of slope angle | ↑ | ↑ | ↑ | ↓ | ↑ | → | ↑ | ↓ | → | ↓ | ↓ | → | ↓ | → | → | ↓ | → | → | ↓ | → |
| Change of erosion rate | → | ↑ | ↓ | ↓ | → | ↓ | ↓ | ↓ | ↓ | ↓ | ↓ | → | ↓ | ↓ | ↓ | ↓ | → | → | → | → |
| Activity today | ++ | ++ | + | + | ++ | ++ | ++ | ++ | + | + | + | 0 | 0 | 0 | ++ | + | 0 | 0 | + | 0 |
| Years since deglaciation | 59 | 66 | 66 | 66 | 77 | 78 | 87 | 88 | 90 | 101 | 108 | 117 | 119 | 120 | 120 | 121 | 123 | 135 | 148 | 154 |

## 5  Conclusion

The following conclusion can be drawn out of our analyses:

1. The use of data covering several decades, which is possible with the processing of historical aerial images, makes a
   space-for-time substitution dispensable. Instead of comparing different moraine sections regarding their different time
   since deglaciation, it is possible to analyse morphodynamics over time on the same moraine section. This enables
   new insights and reveals the true complexity of the development of lateral moraine slopes in glacier forefields. It





shows e.g. that the stage of development of moraine sections which have been deglaciated a longer time ago does not necessarily provide information about the future development of younger moraine sections.

2.  Melting dead ice is an important factor to consider when analysing the development of glacier forefields and especially moraine slopes. This melting ice can lead to an ongoing destabilization of the slopes and thus decelerate the process of stabilization.

3.  The highest morphodynamics occur on very steep moraine slopes with deeply incised gullies. On these slopes, high erosion rates could be detected over many decades, often with ongoing gully incision and headward retreat. Slopes that did not experience a heavy gully erosion and incision seem to show more signs of stabilization.

4.  The time since deglaciation alone does not explain the pattern of active or less active areas within the glacier forefields.

5.  The slope angle seems to be the crucial factor determining the geomorphic activity of moraine slopes. Only on slopes with at least 30° slope angle, high morphodynamics could be detected. The distribution of the glacier ice mass during the LIA maximum as well as historical photographs suggest that from the beginning of deglaciation on, the slope angle on lateral moraines was lower with smaller distance to the LIA maximum.


*Data availability*

The data used in this study are accessible upon request by contacting Sarah Betz-Nutz (sarah.betz@ku.de).

*Author contribution*

SBN conceptualized the study together with MB and FH. SBN conducted the fieldwork, processed and analysed the data. MB, TH and FH supervised the analyses. SBN wrote the draft of the manuscript which was reviewed and edited by MB, TH and FH.

*Competing interests*

The authors declare that they have no conflict of interest.

*Funding*

We gratefully acknowledge the financial support by the German Academic Scholarship Foundation (scholarship for Sarah Betz-Nutz) and the Catholic University of Eichstätt-Ingolstadt during the data acquisition and the conduction of the analysis.
Moreover, we thank the German Research Foundation (DFG) for the financial support within the project "Sensitivity of High Alpine Geosystems to Climate Change Since 1850 (SEHAG)" (FOR2793) which enabled the acquisition of some of the used datasets.



*Acknowledgements*

We would like to thank the Hydrographic Office of the Agency for Civil Protection, Autonomous Province of Bolzano-South Tyrol, Italy, for providing the aerial images of 2016 for the South-Tyrolean study areas, the Autonomous Province of Trento, Italy, for the aerial images of Vedretta d'Amola, the Bavarian State Office for Survey and Geoinformation (LDBV) for the aerial images in the Höllental and the Office of the Tyrolean government, Department of Geoinformation, Austria, to place aerial images of the 1970s and 2000s of Tyrol at the disposal for several of our study areas. We also acknowledge the help

during field work by the student assistants Verena Croce, Andreas Hirtlreiter, Marina Krauß, Franz Wenkowitsch, Angelika Neumann and Alexander Arnold, and the colleagues Georgia Kahlenberg, Jakob Rom, Kerstin Wegner, Fabian Fleischer und Moritz Altmann.

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
