# Peer review of "Development of the morphodynamics on LIA lateral moraines in ten glacier forefields of the Eastern Alps since the 1950s"

_Earth Surface Dynamics, 2022_

## Referee Comment (RC2)

[revised manuscript text omitted]

---

## Author Comment (AC1)

We thank Oliver Sass (RC1) very much for the detailed feedback, recommendations and critical questions. The suggested revisions which will improve the quality of our manuscript. In the following, the original comments of the reviewer (black) are commented in blue.

The paper presents a highly impressive, huge dataset of paraglacial geomorphological processes at lateral moraines. The multitemporal DGM data derives from archival aerial images reaching back to the 1950s as well as from recent drone imagery (400-1600 images per section) and terrestrial laser scans. Ten glacier forefields were investigated, including 1-3 selected moraine sections in each one. For almost all of the sites, three points in time were available for orthophoto evaluation. On each moraine section between 15 and 79 ground control points were fixed and surveyed by dGPS. These numbers show that a unique dataset is presented that definitely warrants publication in Earth Surface Dynamics.

The data is well presented and the conclusions are very interesting for our understanding of paraglacial adjustment of moraine slopes, and of the validity of a space-for-time substitution approach. Thus I recommend acceptance with **minor revisions**.

L75-80: The aims could be pointed out a bit more precisely.
We add a sentence clarifying the aims.

L190 ff (3.4): I don't know exactly how the 3D block thinning works, but it sounds questionable to just automatically delete points from the point cloud. I doubt that this procedure makes comparability and adjustment of the point clouds any better.
We add an explanation of how the 3D blockthinning we used works.
We used the filter method "mean", where a new point representing the mean value within the defined block is calculated. The horizontal and vertical spacing depends on the density of the point cloud and was selected to be between 0.1 m and 1 m in our case. The blockthinning is necessary for a better adjustment of the point clouds (ICP).

L206 ff (3.5): The section on error estimation should be extended by 1-2 explanatory sentences. That is a bit too little information.
We add more information about the selection of stable areas within the DoDs and explain more about the error calculation and its background.

L251: "the profile with the biggest height difference was determined" – why this? That doesn't become clear...
The gully with the biggest height difference between the ridge and the thalweg is the most/deepest incised gully, so it shows the maximum depth of the gullies on the respective moraine slope section. We add that this represents the deepest incised gully.

L257: I tried hard but I do not understand why "the area between the lines was divided by the width of the gullied moraine section". Maybe a small sketch would help.
We add the following sketch:

[Figure]

— Area of headcut retreat between 1959 and 2006  — Moraine section LTF2

Furthermore, we add some information about the calculation of the mean headcut retreat.

L266: "For moraine sections which show so little erosion that no erosion area can be clearly defined, the entire moraine section is considered as erosion area." – I understand the idea behind it; nevertheless, this sounds a bit odd and you might bring a bias into the analysis. When you narrow down active sites to the actual active area, the process rates will inevitably increase. When you don't do the same for less active areas, their rates will be even lower than already. Consider to use the entire slope area in all cases, or give clear reasons why you don't.

We see your point. The problem with considering the entire slope area in all cases is that then the different sizes of the erosion areas compared to the whole slope area determine the erosion rates. As this proportion differs significantly between the moraine sections, that would bring an even higher bias to the analysis.
We suggest to add a table containing the proportion of the area considered as erosion area on the whole moraine section for each moraine section in order to enable better comparability and consider that aspect in the interpretation.
The entire moraine sections were only considered for two moraine sections, APF1 and RKF1. We now define erosion areas also for these two moraine sections, which cover the upper moraine slopes, in a proportion which fits well with the erosion areas of the other moraine sections. The final erosion rates do not change mentionable (only few millimetres) as the area within which we sum up the negative values is also smaller now.

L275: "multiplied by the cell size": Maybe I missed it, but I think the cell size has not been explained nor quantified at this point.
In the line you mention (section 3.6.3), we add a reference to section 3.4 where the cell sizes are mentioned.

L310: "Both parameters are highest for the DoDs based on aerial images."  Is this a problem of the method (likely), or could in be that the amount of surface change was in fact higher in the "arial image" period?
We strongly suppose that this is because of the common quality (angle of perspective and distortions in the images, flight altitude etc.) and the resolution of the aerial images. We see big differences between the different aerial image datasets, which also leads to the high variance of the mean values and standard deviations shown in Fig. 3. In general, the older images are worse and get better over time. However, there are also big differences between the Austrian images and the Italian ones regarding quality and resolution. This is highly influencing on the results of the matching algorithms and thus also on the quality of the resulting DEMs and DoDs.

L311 ff: Deviations of up to 76.5 cm are a bit much for "stable surfaces". I understand that this is an outlier; however, the deviations are just shown, and there is no explanation of how this affects the interpretation. Add a sentence on that.

Yes, we are aware that this is a high error for a stable area. This is due to the poor quality of the used datasets which causes a "salt-and-pepper-pattern", with high deviations in some parts of the model. Regarding the interpretation, we took care when comparing erosion rates with low magnitude (high magnitude changes exceed the error clearly) or moraine sections with little difference between their erosion rates and did not put too much interpretation and meaning on these differences. We add a sentence on that.

L 348: "geomorphic activity beyond the subsidence is low" – Beyound the subsidence - probably yes. But the subsidence itself is, quantitatively, a very important process. It might be triggered by ice melt but this process has set the whole slope in motion and so it shouldn't be discarded as irrelevant dead ice melt. Maybe the entire section should be treated as 4.3.1: Geomorphic processes triggered by dead ice melt. Otherwise it appears to the reader that processes triggered by dead ice melt do not count towards morphodynamics. Later in the paper this becomes clearer, but here it is somewhat misleading.

With the sentence "We infer from the existence of vegetation (Fig. 5b) that geomorphic activity beyond the subsidence is low" we did not want to say that the dead ice melt is irrelevant. On the contrary, it is highly relevant for the slope, because it causes its sliding and subsidence. However, we want to differentiate clearly between ice melt and sediment movement itself (erosion, transport, deposition), especially for the calculation of the erosion rates. In this case, presumably a substantial amount of the "moved" or better disappeared volume is the melted ice and not sediment. This is also visible due to the more reddish than blueish colour in Fig. 5a. A separation is impossible in this case. So, we do not account dead ice or glacier melting itself as morphodynamics, but of course as soon as sediment is moved (e.g. subsidence), possibly induced by dead ice or glacier melting, this is morphodynamics and can be considered as a geomorphic process.

We clarify in section 4.2, wherever possible, that e.g. subsidence or sliding induced by melting dead ice are geomorphic processes.

What we meant with "beyond the subsidence" is that we do not see any geomorphic processes directly on the surface, so processes not induced by the ice melting. We will substitute "geomorphic activity beyond the subsidence" by "no geomorphic processes occur directly on the surface".

We suppose you mean section "4.2 Occurrence of dead ice" should be renamed? We think that is a good idea and substitute the sections' name as you suggest.

L395 ff ("On the DoD 1973–2009...") - Start with pointing out that the prominent red area in Fig. 7b is glacier melt! (You know it, but the reader doesn't...)

We add this aspect.

L405: "ground moraine of an earlier glacier flowing eastward" – Odd statement - doesn't the current glacier also flow eastward? Better: "... from an earlier, larger glacier extent."

The Alpeiner Ferner, so the main glacier tongue, flows mainly northwards (north-north-east), and the former glacier came from the west and joined the main glacier tongue roughly with a 90° angle. We clarify that.

L417 (caption Fig. 7):  - Indicate top and bottom (it is quite clear when you look at it a bit longer, but you could help the reader);  - indicate the glacier extent (maybe with a blue dotted line);  - mention the magenta-coloured polygon (bedrock outcrop in g and h) in the caption

We add these aspects.

L437 (caption Table 4): Headcut retreat in which time period?

We add this information. It refers to the retreat between 1953/1954/1959 (oldest available model) and 2018/2019 (newest available model).

L502: what does "colour map batlow10 from Crameri et al., 2020" mean? Does a colour chart have to be cited?

The colour map (values of the colours) is taken from that author (linked on the ESurf Homepage, colourblind-friendly) and it is written that it should be cited like that. Also, the journal homepage writes "Please cite this source when using the package".

L518 (Fig. 10): I feel that the presentation of the data in this figure is a bit awkward. By summarising the data into just three groups, information is lost. Why not present a scatterplot with time on the x axis and erosion rate on the y axis, using different point signatures for deeply gullied and less deeply gullied sections?

We replace the original plot with a scatterplot and adapt the text referring to this figure.

[Figure]

L562: "at different distances from the LIA maximum" – The result is understandable but I feel that the term "distance from the LIA maximum" is misleading. This could as well be downvalley from the LIA moraines. I assume that "100% distance from the LIA maximum" means "at the current glacier margin"? If yes, write it, e.g. "at different position between recent glacier terminus and maximum LIA extent".

100% distance from the LIA maximum corresponds to the upper border of the defined study area (see beginning of section 2), so where the 1950s glacier extent is still enclosed for a bit (which is at lower altitude than the recent position of the glacier tongues). We adopt your suggestion and adapt it.

L584/585: "… show a decrease of the erosion rates" – No, of the slope angle! (or am I wrong?)

You're right, we correct that.

L590 (Fig. 14): Phew, it took me a while to understand what is depicted here. Is there a simpler way to express this?
We change the x- and y-axis in order to have the independent variable (slope angle 1950) on the x-axis and the dependent variable (change of the slope angle) on the y-axis and hope it is better to understand now.

[Figure]

*Editorial comments*

L83: better "down" instead of "up"
We change that.

L87: delete "mainly the central Alps and the northern and southern Alps" as these sum up to the entire Eastern Alps
We change that

L105: at different distances
We change that.

L132: delete first sentence (because it is meaningless)
We change that.

L288 (Table 3): When the columns DoD1 and DoD2 are identical, it might be better to merge the respective cells.
We change that.

L290: "For the investigation of possible factors influencing the morphodynamics on lateral moraines, besides the detailed analysis of the moraine sections, we conducted also an analysis including the entire defined glacier forefields." Re-arrange this complicated sentence: "We conducted an analysis including the entire defined glacier forefields in order to investigate possible factors influencing the morphodynamics on lateral moraines."
We change that.

L291: "This analysis is based on the negative raster cells of the DoDs and parameters derived from the DEMs." delete – repetition
We change that.

L294: "The most recent DEMs of the entire glacier forefields, so from the 2000s …" Hard to understand, please reword
We change that.

L295: "The resolution of all models is 1 m." This sentence seems to be out of place here, perhaps bring it earlier.
The resolution was set to 1m only for this analysis for reasons of better comparability. Therefore, the information needs to be given in section 3.7. We clarified this with an extension of the sentence.

L389: Delete "gully" at the end of the line
We change that.

L390: "For the gully formation since the 1950s, the moraine section APF3 in the forefield of the Alpeiner Ferner serves as an example" – rearrange to "The moraine section APF3 in the forefield of the Alpeiner Ferner serves as an example of gully formation since the 1950s"
We change that.

L450: "time and again" instead of "again"
We do not know if this happens again and again or if the vegetation cover is constantly destroyed since some years/decades…

L453: show signs of stabilization
We change that.

---

## Author Comment (AC2)

We thank the anonymous referee #2 for the useful comments that helped us improve the manuscript. In the following, the original comments of the reviewer (black) are commented in blue.

The manuscript presents remote sensing epochs of several lateral moraine slopes, and analyses the morphodynamics at each site. It is well-presented, with clear figures and understandable text.

Main Comment:

It would be good to have a slightly longer discussion if possible. Most of the paper focusses on methods, which, although important, does not lend to as significant a contribution to the field. Tieing what the authors analysed and extracted from the data to a broader discussion of morphodynamics and even stresses in slopes (stress paths) would strengthen the paper. The authors can certainly highlight the high resolution (spatial and temporal) of their analysis more, and expand on interpretations. A clearer link between observations and potential geomorphic processes, considering all options, is needed.

The main discussion is included within the results sections, so it is not clear for which aspects a longer discussion is requested. Regarding the relative length of the methods section, we cannot agree with the reviewer as the "results and discussion" section contains about 18 pages in comparison to 7 pages of the methods part. Section 4.5 is discussing all analysed aspects regarding morphodynamics and puts them in the context of the paraglacial adjustment.
Regarding stress paths, this is not part of our study. Geotechnical material properties have not been investigated, and we would like to keep the scope of this work to the long-term morphodynamics observable by means of multitemporal DEMs. The "stress path" suggestion might be valuable for future work though.
The high spatial and temporal resolution of the data used is already highlighted in the introduction (L75 to 79) and the conclusions (L646-651). Moreover, we added a sentence in section 4.5 (L634): "This shows the importance of the analysis of multitemporal DoDs in several glacier forefields in order to properly investigate trends of morphodynamics."
Section 4.3 is showing and discussing the geomorphological processes that were observed on the orthophotos, in the DoDs and during field work, so observations and potential processes are linked.

See attached for further comments.

L27-28: In the European Alps
We continue with the citation of Hambrey:
Hambrey (1994, p.142) wrote that "lateral moraines are among the most impressive features of contemporary glacial mountain environments **[…], especially above and down stream of those glaciers in the Alps, Scandinavia, the Western Cordillera of North America, and elsewhere** […]".

L31: variably consolidated? Some moraines are quite well consolidated. Also, see end of the paragraph - overconsolidation.

Yes, some lateral moraine sections are overconsolidated, whereas the glacigenic sediment in the forefields is in large parts unconsolidated, as described in literature. We deleted "unconsolidated" as it referred only to the moraines in our case.
Moreover, in L32 we added "Curry et al. 2006" and "Chiarle et al. 2007" in the citation.

L83-85: "On the upper border perpendicular to the glacier flow direction, the areas were usually delimited in a way that they include areas still covered by the glacier on the earliest aerial images in the 1950s" →Suggest rephrasing - a bit confusing the way it is currently written.
We rewrote the sentence: "At the upper border, perpendicular to the glacier flow direction, the areas were usually delineated so that they include areas that were still covered by the glacier on the earliest aerial images in the 1950s."

L134: Selected instead of available
That were the datasets which have been available for this study. For some areas, more datasets are generally available (can be bought), but not for all.
We added "for this study".

L167-168: "The mounted camera of the drone was positioned in that way that view direction was orthogonal to the slope surface and for that purpose it was also adapted during the flight. →Did you use terrain-following?
No, we did not use terrain-following. This was not available for the drone we used for this study.

L172: Delete "the"
We deleted "the".

L221: "of" instead of about.
We replaced "about" by "of".

L302: "In a next step, we used two distance grids of subsequent glacier extents to calculate for each pixel the year of deglaciation using a linear interpolation of the respective dates of the glacier extents based on the distance ratio."
What do you define as deglaciation in this case? When the toe of the glacier has receded past the upvalley end of your image window? Or when there is no more ice on the moraine slope within the image window?
We added in L 298: "As mentioned before (section 1), morphodynamics are assumed to dependent amongst others on the time since deglaciation. Deglaciation is understood in this study as the complete melting of the glacier at a particular location, but this does not necessarily include the melt-out of dead ice or ice-cored moraines."
Regarding your comment in section 3.7, deglaciation means the year when the glacier has melted at the location of the single pixel. See L299 "[…] to estimate a value for the time since deglaciation for each raster cell".
Regarding the analysis of the deglaciation of the moraine sections (section 4.5.1), we added in L305: "These interpolated times of deglaciation were also used to define the year of complete deglaciation for each moraine section (youngest year when the foot of the slope was free of glacier ice)."

L310: So, these are means of means and SDs for DoDs in each time period?

What are the factors contributing to variance?

-dataset source (airphotos vs. drone)

-areal coverage of each image

-anything else (time period is similar for each epoch, 20-30 years)

No, the boxplots show the distributions of the mean values and standard deviations of the DoDs, not means of means. As a summary, the range of these absolute values is mentioned in the text. We clarified that by referring to "these mean values".

The quality of the aerial images and its influence on the model quality is mentioned in L312 and L315. That is the most important influence and differs from survey to survey.

We added in brackets in L312: "The large differences within this type of DoDs can be explained by the different quality of the aerial images **(e.g. flight height, image resolution)"**

We added a sentence in L317: "These lower errors can be explained by the better quality of the drone images in comparison to the aerial images, especially the older ones."

The covered time period has no influence on the quality parameters of the single DoDs, only the quality of the DEMs that were used.

L319: Delete "of"

We deleted "of".

L333: Add "depth"

We added "depth".

L338-339: You could also discuss how the rugosity of the moraine slope decreases over time (i.e, the moraine is much smoother in 2016 vs. 1959).

That is right, the surface in 2016 is smoother than it was in 1959. That might be either due to the different quality of the models (quality in 1959 much worse) or a smoothing over time. The latter is also supported by a comparison of the orthophotos as the OF of 1959 shows an undulated surface (cf. Betz-Nutz 2021, p 208). This is supposedly caused by sediment-covered dead ice which is unevenly distributed.

We added in L368-369: "The higher surface roughness in 1959 in comparison to 2016 is supposedly caused by sediment-covered dead ice patches undulating the surface as well as the lower DEM quality of 1959."

L351: Delete „of"

We deleted "of".

L369: It would be good to differentiate between rapid and slow landslides, as you have the data. Are the landslide movements episodic, or steady?

Also, can you make any comments about landslide deposits on the glacier? Can you see deposits from upvalley landslides?

There were only very few slides we could detect. Those are only detected within one (more precisely the first, L363) period and are not detectable on subsequent DoDs. We interpret this as an indication that the landslides were fast and not active on longer timescales (and at low velocity).

We have not analysed deposits on the glacier in detail. However, on the orthophotos no deposits of bigger landslides are visible. Moreover, we did not detect any bigger landslides coming from the adjacent rock slopes to the moraines.

L386: except for instead of except of
We replaced "of" by "for"

L415: Could you calculate volumes of material moving downslope?
The erosion rates are shown in section 4.4.
We added in L391 "Erosion rates are presented in section 4.4."
We refrained from calculating erosion volumes as these depend on the size of the respective erosion areas and impede the comparison of the different moraine sections.

L416-417: It would be helpful to indicate glacier flow direction if possible, just to make it more obvious that the glacier flows roughly S to N, and is retreating to the SSW.
We indicated that and adjusted the figure caption accordingly.

L479: erosion rate and L480: they have
We rephrased the sentence.

L488: Why? Did you also try the volume method?
We did not try to estimate the eroded volumes by reconstructing the gully volume, as this is no accurate method to calculate erosion rates, for example because we also expect the crests between the gullies to lower or even break down with time. Moreover, a development of the intensity of the erosion cannot be analysed over different time periods, as only a general estimation over time can be done with that method. Furthermore, not all moraine slopes are gullied and on the other slopes, no volume can be reconstructed with that method.
Finally, it was the aim of this study to use modern methods such as digital photogrammetry and drone images in order to analyse changes over time at a high resolution.

L520: Perhaps mention here again that year of deglaciation is interpolated. Also, see comment about how you define deglaciation above. The definition is important for considering the stress path the slope material takes to failure/erosion.
We added a comment in L508: "(interpolation method for years of deglaciation see section 3.7)".
Regarding the second part of the section (referring to Fig. 11), from L522 on, "an analysis based on all grid cells" implies that we mean the year of deglaciation for each single grid cell (as described in L298-305, section 3.7, cross reference already given in L523).
The time of deglaciation is important for understanding the geomorphic processes on the slopes. However, we do not consider the stress paths, since this was not the focus of the present work (no geotechnical measurements etc.). We like to stay focused on the detection and quantification of surface changes based on multitemporal elevation models. The "stress path" suggestion might be valuable for future work though.

L532: Why is this section so active?
We cannot explain that fully, however, different possible reasons are given in the paper (mainly in section 4.3.2): recently deglaciated, steep slopes, headward retreat (enough material available), maybe also the still melting dead ice.

L532: Also, consider showing the maximum height of the ice on each moraine slope, if possible.

The maximum ice thickness could in fact be estimated based on the LIA ice extent, but given the length and scope of the present manuscript, we refrain from adding this analysis.

L539: This is not necessarily true. It looks as though erosion rate increases and then decreases through time. Consider testing if this trend is significant.
We described in L525-529 the course of the height difference, in the same way as you write. However, what we would expect is higher erosion rates in the classes with less years since deglaciation. That cannot be supported by the data.
We rewrote in L539: "A comparison of the results of the analysis for the investigated moraine sections (Fig. 10) and the grid cells in the glacier forefields (Fig. 11) shows that in the former lower erosion rates are measured on moraine sections that were deglaciated at earlier points in time, whereas in the latter a consistent decrease of height differences with more years since deglaciation cannot be observed."

L555: Were there any slopes >70deg?
There were only very few single cells >70°, and so these were excluded.
We added in L550: […] 60-70° (single cells with more than 70° were excluded).

L576: Have there been any glacier readvances since LIA? Worth noting for individual cases.
We know there have been smaller readvances in some areas. During these readvances only the glacier tongue was pushed forward, and the glaciers did not cover substantial parts of the slopes again. So, a significant influence on the slope morphodynamics is not assumed. Moreover, we never detected readvances that lasted from one observation period to the next (they were of comparably short duration and we only know that from data which is not presented here) and for some study areas, we do not know anything about readvances as the temporal resolution is either not high enough or there weren't any.

L601: What about (pre-glacial) mountain relief? Mountain slopes generally increase upvalley and toward peaks. Which other factors could explain the slope angle/distance from LIA maximum trend?
We only looked at LIA lateral moraine slopes. The pre-glacial mountain relief can have a certain influence on the distribution of the ice masses of the glacier and with that on the form of the moraine slopes, as explained in L599-601 ("[…], unless certain conditions of the relief, such as larger bedrock outcrops, lead to a congestion of the ice masses also near the LIA maximum (which can be observed e.g. in the Langtauferer forefield)"). As the glaciers' ice thickness (influenced by the relief) explains the slope angles of the moraines very well, we assume that other potentially contributing factors only have minor influence.

L636-637: What about other complicating factors such as glacial readvances, faulting, variable uplift rates, vegetation, etc.?
Regarding the glacial readvances, see the answer above.
Faulting or uplift rates do not seem to be relevant issues for the morphodynamics on lateral moraines as we suppose that the study areas are lifted as a whole. In our DoDs covering the glacier forefields, we could not detect any deviations in altitude that could not be explained by geomorphic processes or that were systematic.
The factor vegetation can have an influence on the stabilization of moraine slopes. We added a sentence in L638: "A stabilizing effect of vegetation growth due to biogeomorphic

interactions is assumed, but the vegetation succession depends on the altitude a.s.l., moisture availability etc. (Betz-Nutz, 2021; Schumann et al., 2016; Eichel et al., 2016)."